# X-chromosome and kidney function: evidence from a multi-trait genetic analysis of 908,697 individuals reveals sex-specific and sex-differential findings in genes regulated by androgen response elements

X-chromosomal genetic variants are understudied but can yield valuable insights into sexually dimorphic human traits and diseases. We performed a sex-stratified cross-ancestry X-chromosome-wide association meta-analysis of seven kidney-related traits ($n = 908,697$), identifying 23 loci genome-wide significantly associated with two of the traits: 7 for uric acid and 16 for estimated glomerular filtration rate (eGFR), including four novel eGFR loci containing the functionally plausible prioritized genes *ACSL4*, *CLDN2*, *TSPAN6* and the female-specific *DRP2*. Further, we identified five novel sex-interactions, comprising male-specific effects at *FAM9B* and *AR/EDA2R*, and three sex-differential findings with larger genetic effect sizes in males at *DCAF12L1* and *MST4* and larger effect sizes in females at *HPRT1*. All prioritized genes in loci showing significant sex-interactions were located next to androgen response elements (ARE). Five ARE genes showed sex-differential expressions. This study contributes new insights into sex-dimorphisms of kidney traits along with new prioritized gene targets for further molecular research.

Chronic kidney disease (CKD) affects about 10% adults globally[1]. By increasing the risk of kidney failure, cardiovascular disease and hospitalization[2], CKD imposes a high economic burden on the healthcare systems[3]. CKD is predicted to become the fifth cause of death by 2040 due to an aging society with increased prevalence of CKD risk factors[4]. Clinical options to prevent, treat or ameliorate CKD are still limited as are CKD randomized controlled trials[5]. A peculiar characteristic of CKD is its sexual dimorphism, with higher prevalence in women but faster progression in men[6]. Investigating the genetic basis of CKD defining traits and kidney function markers accounting for its sexual dimorphism is important to identify molecular targets for tailored pharmaceutical and non-pharmaceutical solutions.

Thus far, hundreds of loci have been identified by genome-wide association studies (GWAS) of kidney function related traits[7–12], extending the overall understanding of the biological basis of CKD and related conditions. However, these studies were mainly limited to autosomal variants and did not consider sex stratification. As for many other common traits, also for CKD-defining traits X chromosome variants are understudied although sexually dimorphic genetic features are more likely to be identified on this chromosome given the differential genetic makeup in males and females. Reasons include analytical challenges due to the differential number of X chromosome copies as well as the X-inactivation in females. Some recent GWAS that included the X chromosome in the analysis unraveled several loci,

e-mail: markus.scholz@imise.uni-leipzig.de

however sex-differential effects received limited attention[13–15]. Since hormones act as transcription factors, hormone response elements in the genome such as androgen response elements (ARE) could provide functional explanations of genetic sex interactions[16]. Indeed, a causal relationship between testosterone and CKD was found in men only[17].

Here, we conducted a cross-ancestry X chromosome-wide association meta-analysis pooling results of 40 studies on up to 908,697 individuals (up to 757,070 European, 152,793 Asian, and 26,371 African ancestry individuals, depending on the trait). We investigated four kidney function markers and three related diseases while accounting for sex-specificity. We identified 23 loci, four of which were not yet described in relation to kidney traits. By means of statistical fine-mapping, colocalization and a comprehensive bioinformatic annotation effort, we prioritized the most likely genes within each locus and identified potential functional consequences. Emphasis was placed on between-trait comparisons and on the analysis of sex-differential effects. For the main CKD-defining trait creatinine-based estimated glomerular filtration rate (eGFR), we identified male-specific effects at *FAM9B* and *EDA2R/AR* and female-specific effects at *DRP2*, along with three sex-differential loci at *DCAF12L1* (larger effect in males), *MST4* (larger effect in males), and *HPRT1* (larger effect in females). ARE are predicted for all of these genes and some also showed sex-biased gene-expression providing functional evidence that could explain the sex-specific and sex-differential findings.

## Results

### Cross-ancestry X chromosome-wide association study

We conducted overall and sex-stratified fixed-effect meta-analyses of X chromosome-wide association scans of seven kidney-related traits and diseases from 40 mainly population-based study groups totaling up to 908,697 individuals with a mean age of 55.7 years (Supplementary Data 1 and 2). Specifically, we analyzed eGFR ($n = 773,980$, mean =

91.33 ml/min/1.73 m²), uric acid (UA; $n = 710,704$, mean = 5.09 mg/dl), urinary albumin-to-creatinine ratio (UACR; $n = 455,053$, mean = 9.65 mg/g), blood urea nitrogen (BUN; $n = 180,748$, mean = 15.05 mg/dl), CKD ($n = 908,697$, including 40,785 cases), microalbuminuria (MA; $n = 517,768$, 36,578 cases), and gout ($n = 195,018$, 2412 cases). Sex ratios were roughly balanced for all traits (45–59% female, Supplementary Data 3). About 80% of study participants were of European ancestry.

After processing, up to 271,730 high-quality single nucleotide polymorphisms (SNPs; Supplementary Data 3), in the overall analysis we identified 14 independent loci significantly associated with eGFR and seven independent loci significantly associated with UA (Fig. 1; Table 1). None of the other phenotypes showed genome-wide significant associations. QQ plots revealed no signs of genomic inflation (Supplementary Fig. 1). Regional association plots of all loci are provided as Supplementary Fig. 2. The index variants at the identified loci explained 0.13% and 0.066% of the eGFR and UA variability, respectively. Heritability of both traits attributable to X-chromosomal variants was estimated and compared between sexes. Estimates for males were significantly larger (eGFR: 0.95% vs. 0.44%, $p = 2.8 \times 10^{-7}$, UA: 0.59% vs. 0.40%, $p = 0.031$, see Supplementary Fig. 3).

In the HUNT study ($N = 69,389$), which was used for validation of the 14 loci associated with eGFR, effect directions were consistent for all index variants and effect sizes were in good agreement (Supplementary Fig. 4, Pearson's $r = 0.96$, $p = 1.1 \times 10^{-8}$). Ten loci showed nominally significant effects in the HUNT study in accordance with the expected statistical power (Supplementary Data 9). The variants explained 0.15% of the eGFR variance in HUNT, a value similar to that found in our meta-analysis.

### Sex-stratified analysis

Sex-stratified analyses revealed an additional genome-wide significant locus for eGFR in males at Xq12 ($p = 3.8 \times 10^{-8}$; Table 1), bringing the

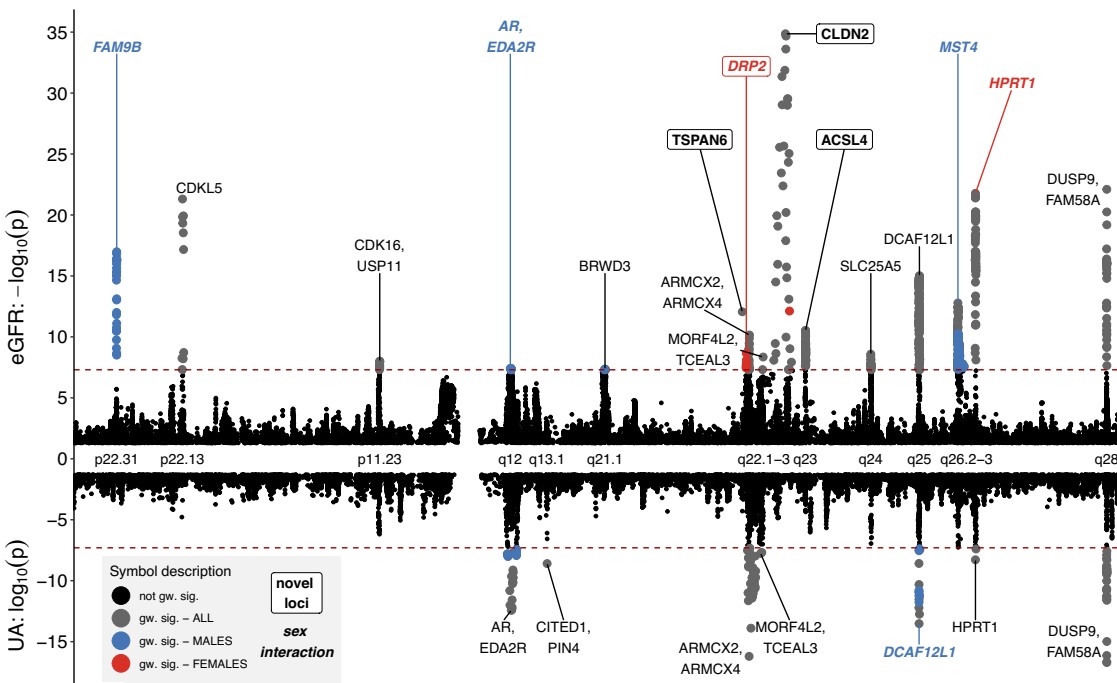

**Fig. 1 | Miami-Plot of variants associated with eGFR and UA.** Results of the cross-ancestry X chromosome-wide association analysis for eGFR (positive panel) and UA (negative panel). Chromosomal position and cytobands of genome-wide significant associations are indicated on the X-axis. The y-axis reports the (negative) $\log_{10}$(p-values) of associations (β coefficient of additive genetic effect in linear regression analysis, two-sided). Values in [−1.3,1.3] are not displayed. Color coding indicate the strata where the smallest *P*-value was observed: overall = gray; male = blue; female = red; and black = not genome-wide significant. Horizontal dashed lines represent the genome-wide significance threshold ($\alpha = 5 \times 10^{-8}$), correcting for multiple testing. Each locus is characterized by candidate gene name, novelty and sex interaction. Bold italics gene names indicate loci with sex interactions and are again color-coded according to the sex with the higher genetic effect size. Novel loci are marked by a box.

**Table 1 | Results of the cross-ancestry X chromosome-wide association analysis of eGFR and UA**

| Locus Number | Index variant | Cytoband/Base Position | Best association | # independent SNPs | Effect/Other Allele | Effect Allele Frequency | Beta (SE) | P value | Phenotypic variance explained (%) | P value Sex Interaction | Candidate genes |
|---|---|---|---|---|---|---|---|---|---|---|---|
| 1 | rs139036121 | Xp22.31 / 8,912,628 | eGFR-Male | 1 | T/C | 0.27 | -0.0032 (0.00037) | $1.0 \times 10^{-17}$ | 0.024 | $2.0 \times 10^{-7}$ | *FAM9B* |
| 2 | rs5909184 | Xp22.13 / 18,482,665 | eGFR-All | 1 | T/C | 0.34 | 0.0027 (0.00028) | $4.9 \times 10^{-22}$ | 0.013 | 0.36 | *CDKL5* |
| 3 | rs72616719 | Xp11.23 / 47,166,532 | eGFR-All | 1 | A/G | 0.58 | -0.0015 (0.00026) | $9.9 \times 10^{-9}$ | 0.0044 | 0.59 | *CDK16, USP11* |
| 4 | rs189618857 | Xq12 / 66,156,010 | eGFR-Male | 1 | A/T | 0.83 | 0.003 (0.00055) | $3.8 \times 10^{-8}$ | 0.013 | $1.5 \times 10^{-3}$ | *AR, EDA2R* |
| 5 | rs2063579 | Xq21.1 / 79,925,246 | eGFR-All | 1 | G/A | 0.7 | -0.0015 (0.00028) | $4.3 \times 10^{-8}$ | 0.0039 | 0.14 | *BRWD3* |
| 6 | **rs1802288** | Xq22.1 / 99,890,204 | eGFR-All | 1 | T/C | 0.17 | -0.0028 (0.00038) | $8.7 \times 10^{-13}$ | 0.0084 | 0.065 | ***TSPAN6*** |
| 7Aᵃ | rs3850318 | Xq22.1/100,938,892 | eGFR-All | 1 | C/G | 0.62 | -0.002 (0.00031) | $7.0 \times 10^{-11}$ | 0.0057 | 0.85 | *ARMCX2, ARMCX4* |
| 7B | **rs149995096** | Xq22.1 / 100,479,327 | eGFR-Female | 1 | T/C | 0.18 | -0.0031 (0.00052) | $2.2 \times 10^{-9}$ | 0.0093 | $5.1 \times 10^{-4}$ | ***DRP2*** |
| 8 | rs11092455 | Xq22.2 / 102,925,716 | eGFR-All | 1 | T/C | 0.44 | 0.0016 (0.00027) | $4.5 \times 10^{-9}$ | 0.0052 | 0.26 | *MORF4L2, TCEAL3* |
| 9 | **rs181497961** | Xq22.3 / 106,168,067 | eGFR-All | 2 | A/G | 0.023 | -0.012 (0.00099) | $1.4 \times 10^{-35}$ | 0.026 | 0.81 | ***CLDN2*** |
| 10 | **rs5942852** | Xq23 / 109,094,393 | eGFR-All | 1 | C/T | 0.76 | -0.002 (0.00031) | $3.0 \times 10^{-11}$ | 0.0058 | 0.42 | ***ACSL4*** |
| 11 | rs16275 | Xq24 / 118,582,383 | eGFR-All | 1 | G/A | 0.68 | -0.0016 (0.00027) | $2.6 \times 10^{-9}$ | 0.0046 | 0.73 | *SLC25A5* |
| 12 | rs5931180 | Xq25 / 125,656,689 | eGFR-All | 1 | A/T | 0.37 | -0.0021 (0.00026) | $9.8 \times 10^{-16}$ | 0.0084 | 0.38 | *DCAF12L1* |
| 13 | rs5933079 | Xq26.2 / 131,251,326 | eGFR-All | 1 | T/C | 0.26 | 0.0022 (0.0003) | $1.7 \times 10^{-13}$ | 0.0071 | 0.025 | *MST4* |
| 14 | rs5933443 | Xq26.1 / 133,797,249 | eGFR-All | 1 | A/T | 0.68 | 0.0029 (0.0003) | $1.7 \times 10^{-22}$ | 0.013 | 0.047 | *HPRT1* |
| 15 | chr23:152898260 | Xq28 / 152,898,260 | eGFR-All | 1 | A/C | 0.54 | -0.0026 (0.00027) | $7.9 \times 10^{-23}$ | 0.013 | 0.97 | *DUSP9, FAM58A* |
| 16 | rs6625094 | Xq12 / 66,301,811 | UA-All | 1 | T/G | 0.82 | -0.024(0.0033) | $3.8 \times 10^{-13}$ | 0.01 | 0.13 | *AR, EDA2R* |
| 17 | rs34687188 | Xq13.1 / 71,509,443 | UA-All | 1 | AT/A | 0.72 | 0.017(0.0028) | $2.6 \times 10^{-9}$ | 0.0069 | 0.6 | *CITED1, PIN4* |
| 18 | rs34884874 | Xq22.1/100,885,798 | UA-All | 1 | CT/C | 0.75 | 0.023 (0.0027) | $6.2 \times 10^{-17}$ | 0.014 | 0.36 | *ARMCX2, ARMCX4* |
| 19 | rs34815154 | Xq22.1 / 102,552,032 | UA-All | 1 | A/AT | 0.37 | -0.014 (0.0024) | $1.7 \times 10^{-8}$ | 0.0062 | 0.34 | *MORF4L2, TCEAL3* |
| 20 | rs112708523 | Xq25 / 125,602,218 | UA-All | 1 | A/AT | 0.4 | -0.018 (0.0024) | $3.0 \times 10^{-14}$ | 0.011 | 0.034 | *DCAF12L1* |
| 21 | rs202138804 | Xq26.3 / 133,799,101 | UA-All | 2 | A/AGT | 0.76 | -0.016 (0.0027) | $5.2 \times 10^{-9}$ | 0.0066 | 0.86 | *HPRT1* |
| 22 | rs4328011 | Xq28 / 152,898,261 | UA-All | 2 | A/G | 0.55 | 0.009 (0.001) | $2.0 \times 10^{-17}$ | 0.011 | 0.15 | *DUSP9, FAM58A* |

Listed are genome-wide significant loci on chromosome X associated with eGFR (first 16 rows) and UA (last seven rows). Association statistics and two-sided p-values of β-coefficients of additive genetic effects are provided for the most associated variant in each locus (index variant). We used the threshold of α = 5×10⁻⁸ to correct for multiple testing. Respective analysis group is provided in column 4. Novel loci are presented in bold while novel sex-interactions are underlined. EAF = frequency of the effect allele.
ᵃeGFR locus 7 was split up since secondary analyses revealed two independent loci, one for the overall analysis and one female-specific.

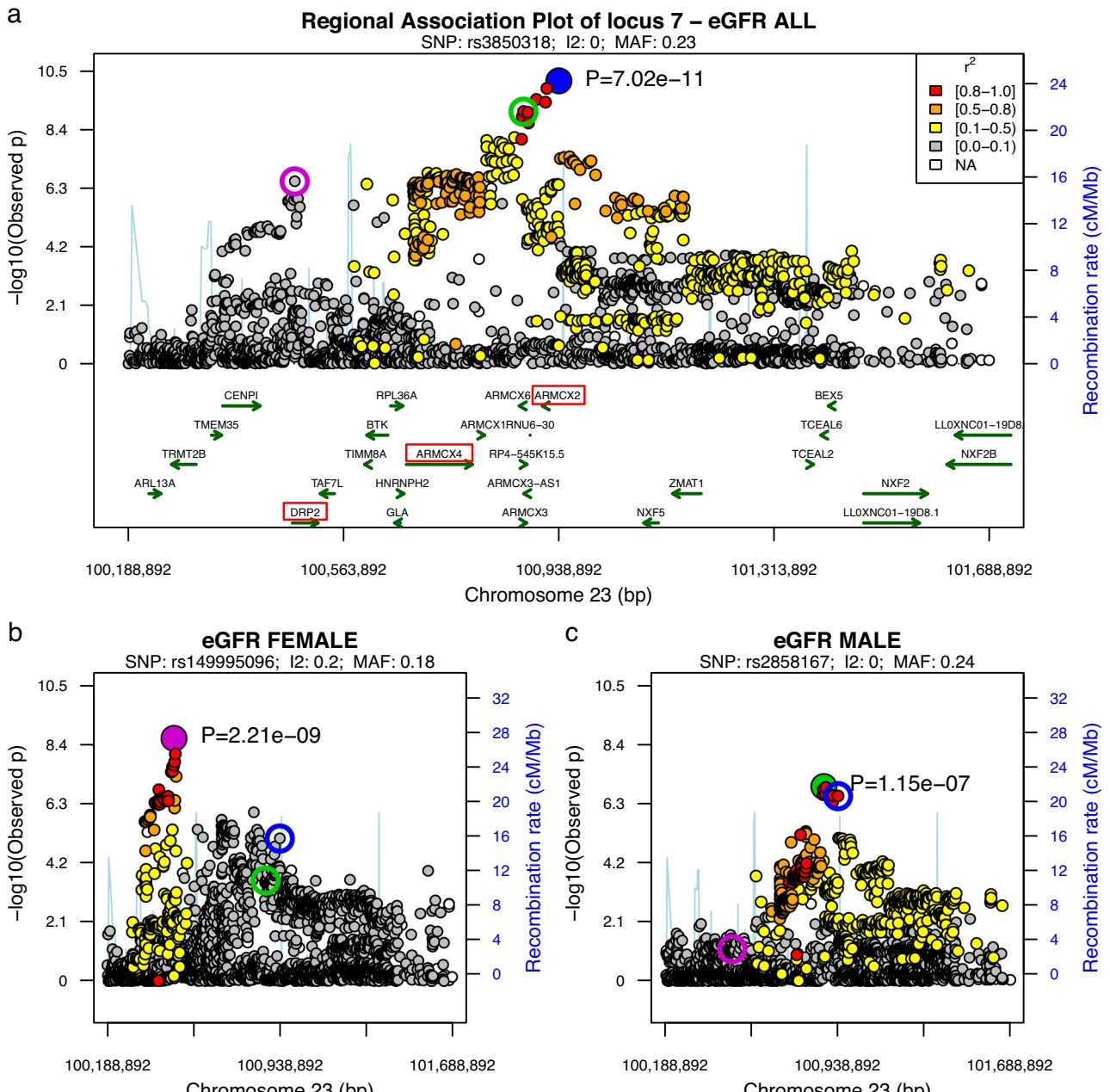

**Fig. 2 | Regional association plot of locus 7 at Xq21.1.** Comparison of association plots of eGFR at locus 7 at Xq21.1 between analysis groups overall, males and females. The same genomic region is displayed for overall (**a**), female (**b**) and male analysis (**c**), respectively. The y-axes report the (negative) $\log_{10}$(p-values) of the associations (β coefficient of additive genetic effect in linear regression analysis, two-sided). The primarily identified index variant rs3850318 showed no sex-dimorphism and is confined to the same haploblock as the best associated SNP in males (**c**). Colocalization analysis revealed an independent association (PP(H3) = 95%), which is significant in females only (rs149995096) and corresponding to a neighboring haploblock (**b**). Subgroup-specific top-hits are depicted as filled circles or marked as empty circles (blue = overall, green = male, magenta = female). In red, we highlight the candidate genes for this locus (**a**).

total number of significant loci to 22. Colocalization analyses of male and female associations at these 22 loci suggested the same causal variant in eight instances (posterior probability PP of H4 ≥ 75% Supplementary Data 4). An effect mainly driven by associations in males was suggested at six loci. Of note, for locus 7 (Xq22.1) detected in the eGFR overall analysis, colocalization analysis suggested different causal variants in males and females (PP(H3) = 95%). At this locus, a detailed analysis revealed a female-specific association with eGFR at variant rs149995096 ($p_{female} = 2.2 \times 10^{-9}$, $p_{male} = 0.071$, $p_{interaction} = 5.1 \times 10^{-4}$), which was not correlated with the index variant rs3850318 of this locus (linkage disequilibrium, LD $r^2 = 0.016$, $p_{female} = 8.2 \times 10^{-6}$, $p_{male} = 2.5 \times 10^{-7}$, $p_{interaction} = 0.85$). The locus is

located at a neighboring haploblock of the signal detected in the overall analysis (Fig. 2). Therefore, we added this hit to our locus list by splitting locus 7 into 7A for the overall hit and 7B for the female-specific hit. Thus, a total of 23 loci were detected (Table 1).

SNP by sex-interaction testing of the 23 index variants identified five nominally significant interactions for eGFR and one for UA (Fig. 3, Supplementary Data 4) including the female-specific finding at locus 7B (Xq22.1). Two variants located in locus 1 (Xp22.31) and 4 (Xq12), respectively, showed significantly larger effects in males while being not significant in females. Accordingly, they were classified as male-specific variants, as further supported by the respective colocalization analyses (Supplementary Data 4).

## Fine-mapping

Conditional analysis revealed a second independent variant at three loci, one for eGFR (locus 9, Xq22.3) and two for UA (loci 21 at Xq26.3 and 22 at Xq28) in the overall analysis. No further independent variants arose from the fine-mapping analysis of the sex-specific subgroups. Annotations of independent variants per analysis can be found in Supplementary Data 6.

## Analysis of heterogeneity between ancestries

Meta-regression analysis accounting for ancestries confirmed the associations observed at all loci (Supplementary Data 11). Only the index variant rs4328011 at locus 22 (Xq28) showed heterogeneous effects on UA across ancestries ($p_{\text{Het-Anc}} = 5.9 \times 10^{-19}$; Supplementary Fig. 5), likely due to allele frequency differences across ancestries (G allele frequency: 48% in African Americans, 42% in Europeans, 62% in Asians according to the ALFA data base of dbSNP version 155). These frequencies align with the data included in the presented meta-analysis.

For UA, meta-regression revealed two further loci with genome-wide significance (Supplementary Fig. 6). One identified by SNP rs57434549 (Xq13.1, $p = 1.6 \times 10^{-12}$, $p_{\text{Het-Anc}} = 1.3 \times 10^{-9}$) showed a positive effect of the T allele on UA in Europeans and Asians, but a negative effect in African Americans. This was also observed within the MVP

study, which contains substantial proportions of participants of European and African American ancestries. Frequencies of the T allele were 19% in Africans, 39% in Europeans, and 51% in Asians, according to ALFA, which is confirmed by our data. The association was also found in ref. 15. The second locus was identified by variant rs1802288 (Xq22.1, $p = 2.4 \times 10^{-8}$, $p_{\text{Het-Anc}} = 1.4 \times 10^{-6}$, frequencies of the T allele: African Americans = 3%, Europeans = 17%, Asians = 0.06%). This SNP was also detected as associated with eGFR in our cross-ancestry meta-analysis (locus 6). Annotations of variants are provided in Supplementary Data 12.

## Comparison of eGFR and UA hits

To investigate whether eGFR and UA loci shared the same underlying variant, we performed LD analysis between index variants and colocalization analyses of overlapping loci. In total, we observed seven physically overlapping loci of eGFR and UA signals comprising eight index variants of eGFR and six index variants of UA (Table 2). To be conservative with claiming different loci, overlap was assumed if either LD $r^2 \geq 0.1$ or PP(H4) $\geq 50\%$. Accordingly, associations at the eGFR loci 4, 7A, 8, 12, 14, and 15 shared the same causal variant with the UA loci 16, 18, 19, 20, 21 and 22. In contrast, colocalization analysis revealed distinct signals for the female-specific eGFR signal at locus 7B and the UA-associated locus 18 (PP(H3) = 99%, $r^2 = 0.02$), and between loci 6

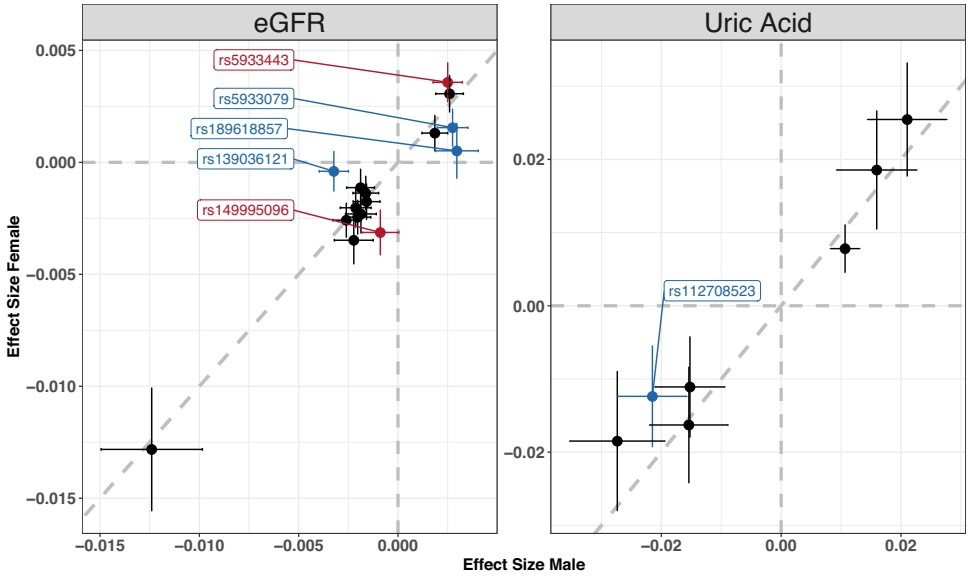

**Fig. 3 | SNP-by-Sex interaction analysis.** We tested all identified 23 index variants for interaction with sex. Comparisons of effect sizes (β coefficient of additive genetic effects) and respective 95% confidence limits between males (x-axis) and females (y-axis) are displayed. Six variants achieved nominal significance (red = higher effect size in females, blue = higher effect size in males).

## Table 2 | Analysis of the overlapping of eGFR and UA loci

| Region eGFR, UA | Cytoband | Index variant eGFR/index variant UA | Best associations | Discordant Effect | LD $r^2$ | PP H3 (%) | PP H4 (%) | Overlap |
|---|---|---|---|---|---|---|---|---|
| 4, 16 | Xq12 | rs189618857/rs6625094 | Male/Overall | Yes | **0.9** | 73 | 27 | Yes |
| 6, 18 | Xq22.1 | rs1802288/rs34884874 | Overall/Overall | Yes | **0** | **75** | 25 | No |
| 7A, 18 | Xq22.1 | rs3850318/rs34884874 | Overall/Overall | Yes | **0.96** | 7 | **93** | Yes |
| 7B, 18 | Xq22.1 | rs149995096/rs34884874 | Female/Overall | Yes | **0.02** | **99** | 0 | No |
| 8, 19 | Xq22.1–2 | rs11092455/rs34815154 | Overall/Overall | Yes | **0.28** | 45 | **54** | Yes |
| 12, 20 | Xq25 | rs5931180/rs112708523 | Overall/Overall | No | **0.98** | 47 | **53** | Yes |
| 14, 21 | Xq26.3 | rs5933443/rs202138804 | Overall/Overall | Yes | **0.99** | 48 | **52** | Yes |
| 15, 22 | Xq28 | chr23:152898260/rs4328011 | Overall/Overall | Yes | **1** | 0 | **1** | Yes |

We analyzed possible overlaps of eGFR and UA loci by comparing best associations of respective index variants. Overlap evaluation is based on linkage disequilibrium (LD) $r^2$ between variants and colocalization results (H3 = no shared signal, H4 = hared signal). PP = posterior probability. To be conservative with claiming different loci, we evaluated $r^2 \geq 0.1$ or PP(H4) $\geq 50\%$ as evidence for overlap. Values relevant for overlap evaluation are displayed in bold. Full colocalization results are provided in Supplementary Data 5c.

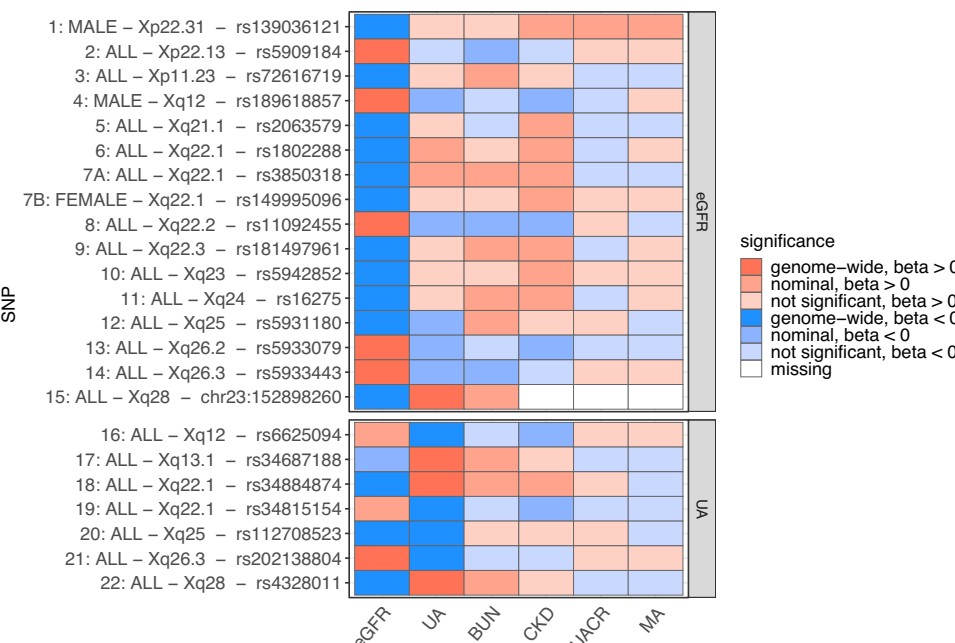

**Fig. 4 | Cross-phenotype comparison of eGFR and UA loci.** We visualized cross-phenotype p-values of association (β coefficient of additive genetic effects) of our 23 index variants showing associations with eGFR or UA to assess potential kidney-related function (BUN), clinical relevance (CKD) and kidney damage (UACR, MA) and to compare eGFR and UA associations. Variants are ordered according to Table 1 and statistics of the best associated analysis group are shown per index variant (see left column). While eGFR and UA are tested two-sided (discovery), associations with the other traits were tested one-sided assuming the opposite effect direction of eGFR for eGFR hits and the same effect direction with UA for UA hits. Of note, effect sizes of eGFR and UA are always opposite except for one locus (locus 12/20, Xq25).

(eGFR overall) and 18 (UA overall) (PP(H3) = 75%, $r^2 = 0$). Of note, effect directions between eGFR and UA were opposite for all overlaps as expected, except for the overlap of loci 12 and 20 (Table 2). UA locus 17 (Xq13.1) showed no overlaps with eGFR loci and, thus, represents the only primary UA locus found in our analyses.

### Cross-phenotype comparisons of loci
Since serum creatinine may also reflect muscle metabolism in addition to kidney function, we analyzed whether eGFR index variants were also associated with BUN. As expected, we observed opposite effect directions for 15 out of the 16 eGFR-related index SNPs. Locus 5, Xq21.1 was the only exception, but the effect was not significant in BUN (Fig. 4). From the 15 opposite effects, nine were nominally significant, classifying them as likely associated with kidney function rather than creatinine metabolism.

To assess clinical relevance, we also checked whether the eGFR index variants were associated with CKD risk. Indeed, this was the case for all index variants except for locus 15 (Xq28), for which no CKD association statistic was available. From the available 15 associations, all were in the opposite effect direction compared to eGFR, 11 were nominally significant and five reached a $p < 0.001$ (Supplementary Data 5a).

We also assessed the relevance of our index variants with respect to kidney damage. Only one of the loci showed significant association with UACR and MA, with opposite effect direction compared to eGFR (locus 1, Xp22.31). Finally, we compared the effect direction of eGFR and UA, and found the expected inverse directions for all but one of the (overlapping) loci (locus 12/20, Xq25; Fig. 4, Supplementary Data 5a, b).

### Replication of previous findings
We investigated genome-wide significant X-chromosomal index variants reported previously[13–15] for associations with kidney-related traits. A total of 46 associations with variants distributed over 19 cytobands were retrieved from these studies. From the 34 associations with

available summary statistics in our study, we successfully replicated all at nominal one-sided significance (Supplementary Data 10).

### Enrichment of ARE genes
According to our gene-prioritization strategy, we assigned candidates with AREs to all of the six loci with sex-interactions. As identified by a simulation study (see methods), this represents an enrichment ($p = 0.014$). Moreover, for two of the loci, genes predicted to be regulated by AREs[18] could be assign, again representing an enrichment ($p = 0.025$).

### Single locus results
Accounting for the locus overlaps between eGFR and UA, 17 non-overlapping loci remained and will be discussed in the following. A total of 13 of these loci where previously described in refs. 13–15. For five of these loci, we discovered new sex-differential findings, while four loci where not yet described, including a sex-specific one (Table 1). We assigned functionally plausible candidate genes to all of the loci, following a gene-prioritization strategy (see methods). Detailed reasoning for selection of candidate genes can be found in the Supplementary Notes 1.

### Known loci
Our gene-prioritization strategy confirmed previously proposed candidate genes for loci 2 (Xp22.13: *CDKL5*) and 15/22 (Xq28: *DUSP9*), and proposed new or additional candidate genes for known loci 3 (Xp11.23: *USP11/CDK16*), 5 (Xq21.1: *BRWD3*), 8/19 (Xq22.2: *MORF4L2, TCEAL3*), 11 (Xq24: *SLC25A5*), 15/22 (Xq28: *FAM58A*) and 17 (Xq13.1, *CITED1, PIN4*), see Fig. 1, Table 1 and the Supplementary Notes 1 for a detailed reasoning.

### Known loci with sex-interactions
We discovered sex-interactions at five previously described loci. Genes with AREs could be assigned as candidate genes for each of them:

**Locus 1 (Xp22.31)**. The strongest association with eGFR was observed for rs139036121 in males. BUN was not significantly associated with this locus. We observed a pronounced sex-interaction with no association in females (Fig. 3). The locus is pleiotropic, with a variety of other GWAS associations including several sex-specific traits such as testosterone levels and male-pattern baldness[19,20]. Moreover, we observed colocalization between eGFR and testosterone associations at this locus in males (PP(H4) = 99%) with opposite effect directions, i.e., the eGFR association could be driven by a primary testosterone effect (Supplementary Data 14). The nearest candidate gene is *FAM9B*, which has an ARE 70 kb upstream of its transcription start side (TSS)[18].

**Locus 4/16 (Xq12)**. The index variant rs189618857 was associated with eGFR only in males, with a strong sex-interaction ($p_{IA} = 1.5 \times 10^{-3}$, Fig. 3). The association with BUN was not significant. The index variant is in LD with the index variant of UA locus 16. Other GWAS traits associated at this locus comprise sex hormone-binding globulin levels, male-pattern baldness, fasting insulin, estradiol levels with the same effect direction and prostate cancer risk. Rs189618857 maps to a gene desert. The credible set (CS) of variants identified within this very large eGFR locus comprised 537 variants, including strong CADD score variants (CADD > 10) near *EDA2R*, a plausible candidate gene encoding the ectodysplasin A2 receptor[21]. LD with expression quantitative trait loci (eQTLs) of *EDA2R* and *AR* were also observed for this locus (Supplementary Data 7). *AR* encodes the androgen receptor, and has upstream estrogen response elements[22]. It therefore constitutes another plausible candidate gene of this locus[23]. An ARE was estimated 5kB upstream of its TSS[18]. *EDA2R* also has an ARE in some distance from the gene body. Both genes were shown to be regulated by the ARE (*AR* up-regulated, *EDA2R* down-regulated[18]). Moreover, *AR* shows significantly higher gene expression in females while *EDA2R* shows higher expression in males in several tissues[24]. Thus, both *EDA2R* and *AR* are plausible candidates here.

**Locus 12/20 (Xq25)**. The index SNP rs5931180 was associated with eGFR in the overall analysis. Relation to kidney function was supported by a significant inverse association with BUN. The variant is in LD with the index variant of UA locus 20 showing significantly larger effects in males (Fig. 3). In contrast to other overlaps of eGFR and UA, we observed the same genetic effect directions for eGFR and UA. The credible set comprised 66 variants for eGFR and 45 variants for UA, with a sharp signal centered on the genes *MTND4P24* and *DCAF12L1*. *DCAF12L1* has an ARE 3 kb downstream (3'UTR) of the TSS and is higher expressed in males in kidney cortex[24], possibly explaining the sex-differential effect. Therefore, it is considered the likely candidate here.

**Locus 13 (Xq26.2)**. The index SNP rs5933079 was most strongly associated with eGFR, with larger effect size in males. CKD and UA but not BUN were associated with opposite effect direction. The credible set contained 39 variants, including variants with strong deleteriousness scores (CADD > 10) near *FRMD7*, *RAP2C* and within *MST4*, respectively. There are AREs near *RAP2C* (50kp upstream) and *MST4* (28kB downstream), while both genes are also found to be down-regulated by their AREs[18]. There is additional kidney-related evidence related to *MST4*[25,26]. Moreover, MST4 was shown to correlate with androgen receptor status in prostate cancer cell lines revealing male-specific functionality[27]. Thus, we propose this gene as the most likely candidate here.

**Locus 14/21 (Xq26.3)**. The strongest association at locus 14 was observed for rs5933443 for eGFR in the overall analysis, with a significant sex-interaction showing larger effects in females, and significant BUN association in the opposite direction. The index variant is

in strong LD with rs202138804 associated with UA. CS variants map to the gene bodies of *PLAC1*, *HPRT1*, *FAM122B*, and *PHF6*, with the strongest CADD scores observed for *PLAC1*. Three of these genes show AREs in some distance (*PLAC1*, *HPRT1*, *FAM122B*), two also show estrogen response elements (*PLAC1*, *HPRT1*) possibly explaining the sex-interaction. *HPRT1* also shows higher expression in females[24]. *HPRT1* encodes hypoxanthine phosphoribosyltransferase, a central enzyme in the generation of purines such as UA. Thus, the biological link to the observed association with UA is closer than the one observed with eGFR. Rare loss-of-function variants in *HPRT1* are a cause of Lesch-Nyhan Syndrome featuring highly elevated levels of UA (OMIM-ID 308000)[28]. In consequence, *HPRT1 is* the most plausible candidate gene at this locus.

### Novel loci
Four loci were not yet described in the literature. Another strong sex-interaction was found for one of them.

**Locus 6 (Xq22.1)**. The locus was most strongly associated with eGFR in the overall analysis (rs1802288). It was also associated with CKD and UA, but not BUN. The association with UA achieved genome-wide significance after adjusting for ancestry with MR-MEGA (Supplementary Fig. 6A, B). The locus was described for association with height. The credible set contained only the index variant with a pronounced CADD score of 29.9. The SNP is a missense mutation in *TSPAN6* (Ala108Thr, Supplementary Fig. 7). A relationship of this gene with kidney function was not yet described. However, of note, another member of the TSPAN family, namely *TSPAN33* located at chromosome 7 was proposed as a candidate gene of eGFR association in the study of ref. [14].

**Locus 7/18 (Xq22.1)**. The strongest association was observed for rs3850318 with eGFR in the overall analysis. The variant was also associated with BUN, CKD and UA, i.e., this association overlaps with locus 18 of UA (rs34884874, colocalization PP(H4) = 93%). Moreover, the index variant is in LD with associations with creatinine and UA reported in Sakaue et al.[15]. We observed co-localizations of the eGFR and the UA signals with an eQTL signal of *ARMCX2* in kidney tubulointerstitial tissue (eGFR: PP(H4) = 82%, opposite effect direction, UA: PP(H4) = 95% same effect direction, Fig. 5, Supplementary Data 8, 15), thus prioritizing this gene.

Since colocalization analysis between male and female eGFR results at this locus strongly supported the hypothesis of different signals (PP(H3) = 95%, Supplementary Data 4), we analyzed this phenomenon in more detail by looking at the sex-stratified results of eGFR. The top-variant in males was rs2858167, which is 62kB away from rs3850318, still the variants are in LD ($r^2 = 0.83$, Fig. 2). The SNP did not achieve genome-wide significance in males and no significant sex-interaction was observed ($p_{IA}=0.32$). Conversely, the top-variant in females was rs149995096, which is 460 kb away from rs3850318 and is not in LD with this variant nor with the male top-hit ($r^2 < 0.018$). Of note, rs149995096 achieved genome-wide significance in females while the effect in males was not even nominally significant ($p_{IA}=5.1 \times 10^{-4}$). Thus, this variant is an independent female-specific hit of this locus. Moreover, this variant was not in LD with other reported GWAS variants, i.e., it represents a novel finding. CKD but not BUN was associated. The variant is in the coding sequence of *DRP2* and the credible set comprising 92 variants also contains variants with high CADD scores within or near this gene. Since the association with BUN was not significant, the eGFR association could also be related to muscle mass. In this regard, *DRP2* could be a plausible candidate due to its relationship to creatinine via involvement in muscle dystrophy[29]. Of note, the gene has an ARE 17 kb downstream of the TSS and shows higher expression in females in several tissues[24]. Since there is no evidence of X-inactivation escape of this gene[30,31], this gene-expression difference is likely caused by different regulation but it is

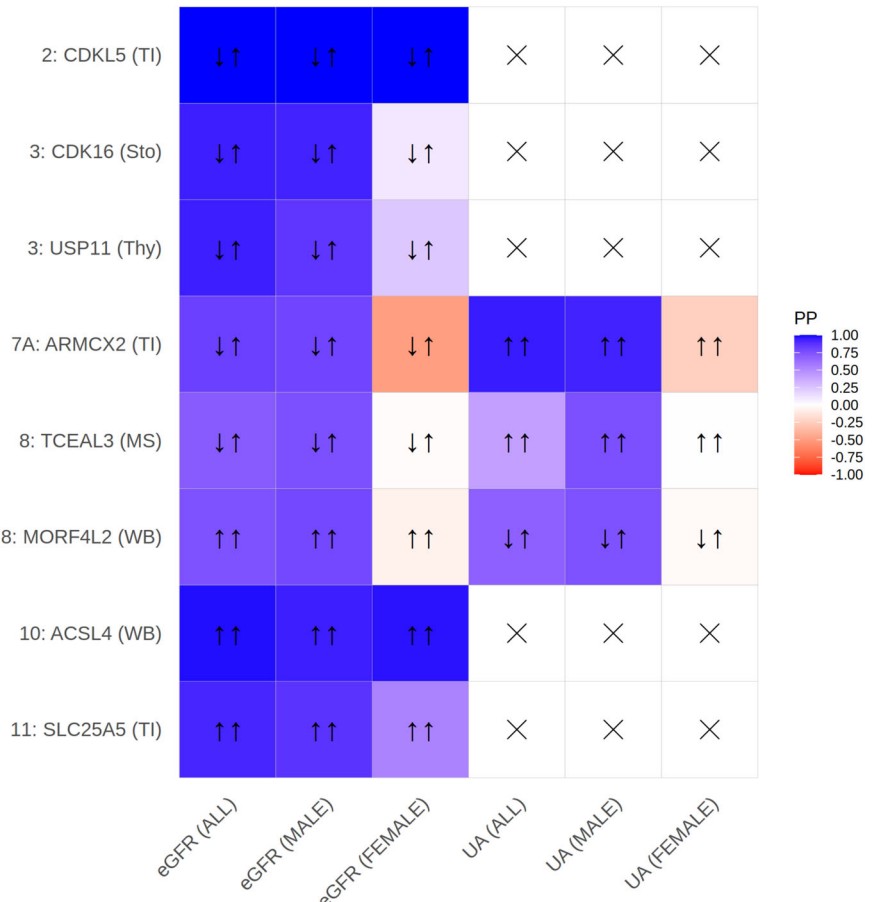

**Fig. 5 | Results of eQTL-colocalization analysis.** We present selected positive colocalization findings of eQTLs and kidney traits. We display genes showing co-localization posterior probabilities (PP) larger than 75% for kidney-related tissues (tubulointerstitial (TI)) or for kidney-related genes in other tissues (stomach (Sto), thyroid (Thy), muscle skeletal (MS), whole blood (WB)) for at least one kidney trait in at least one analysis subgroup. Color coding corresponds to posterior probabilities of hypotheses H3 (different signals for kidney trait and eQTL, red) and H4 (same signal, i.e., colocalization, blue). Arrows show same (↑↑) or opposite (↑↓) directions of effects of trait and eQTL. To assess this relationship, we used the index variant of each locus, respectively for locus 8 the best available proxy (rs12851072, $r^2 = 0.97$). In case of co-localizations in multiple tissues, we showed the results for kidney-related tissues or the tissue with the strongest support for H4. All results can be found in Supplementary Data 8. Direction of effects are provided in Supplementary Data 15. Crosses represent combinations not tested due to lack of signal for the kidney-related trait.

unlikely that this explains the observed eGFR association due to lack of colocalization of gene-expression and eGFR signals at this locus (Supplementary Data 8).

**Locus 9 (Xq22.3).** At this locus, the index variant rs181497961 was associated with eGFR in the overall analysis, without sex-interaction. The variant was also associated with BUN and CKD with opposite effect directions. Ref. 14 reported an independent ($r^2 = 0.093$) eGFR association about 600kB away from this variant, namely rs56121637, which also showed genome-wide significance in our analysis ($p = 8.3 \times 10^{-10}$). Ref. 32 also found this variant to be associated with creatinine. Both groups proposed *RNF128* as the causal gene.

Conditional analysis revealed the presence of another independent variant at this locus, namely rs111410539 not in LD with the variants mentioned above ($r^2 < 0.1$). Due to the small allele frequencies of the variants, the respective credible sets were large, comprising 408 and 1583 variants, respectively. Of note, rs181497961 did not achieve the highest PP of its credible set, which was attributed to rs111775083 with a higher effect allele frequency of 5.6%. No eQTL co-localizations were detected for this locus. The index variant is in the gene body of *MORC4* and *CLDN2*. Although, several high CADD score variants within different gene bodies are in the CS, we consider the gene *CLDN2* as a highly plausible candidate due to its known role in nephrolithiasis development according to OMIM-ID 300520 and the kidney phenotypes of *CLDN2* knock-out mouse models[33].

**Locus 10 (Xq23).** The top-SNP of this locus rs5942852 was best associated with eGFR in the overall analysis. The variant was associated with CKD but not with BUN. No correlated GWAS hits were found, suggesting that this association is a novel finding. The credible set comprises 54 variants. The top-variant is near *RPSSP7* and there is also a high CADD variant nearby (CADD = 12.6). However, this gene has no known functional relationship to kidney traits. Although 120kB away, the locus co-localizes with an eQTL of *ACSL4* with the same effect direction in blood (PP(H4) = 98%, Fig. 5) and other tissues. *ACSL4* also known as *FACL4* could be a plausible candidate since it was linked to Alport syndrome[34].

## Discussion

We performed a cross-ancestry meta-analysis of genetic associations between X-chromosomal variants and seven kidney traits in up to 908,697 individuals. Particular emphasis was placed on sex-stratified analyses to account for the specific nature of X-chromosomal genetics. Moreover, we performed cross-phenotype comparisons of genetic associations to provide a comprehensive characterization of the identified loci with respect to their associations with different kidney traits.

In total, we identified 23 loci, seven for UA and 16 for eGFR. Loci of UA were mostly overlapping with those of eGFR, i.e., only one of the UA loci, namely Xq13.1, showed no association with eGFR. Four of the eGFR loci represented novel findings and established genome-wide significant associations for functionally plausible prioritized genes, namely *ACSL4*, *CLDN2*, *DRP2*, and *TSPAN6*. The *DRP2* locus was female-specific.

Further, we identified novel sex-interactions with genetic variants at five additional, previously described loci, comprising two male-specific (*FAM9B*, *AR/EDA2R*) and three sex-differential findings, two with stronger genetic effects in males (*DCAF12L1*, *MST4*) and one with a stronger effect in females (*HPRT1*). All prioritized genes of these loci contain hormone response elements, in particular AREs, providing possible functional explanations of the sex-specific or sex-differential effects at these loci.

Several lines of evidence suggest that sex hormones may play a role in kidney function and may contribute to sexual dimorphism of CKD. Higher levels of the sex hormone binding globulin (SHBG), a modulator of several sex hormones, have been causally associated with lower CKD risk[35] and gout[36] in men but not in women. Androgens are inversely associated with kidney function in men,[37] with testosterone being causally associated with lower creatinine- and cystatin-based eGFR as well as increased risk of CKD and albuminuria in men[17]. Dihydrotestosterone may lead to dysregulation of several metabolic pathways associated with diabetes and CKD[38]. In contrast, lower estrogen levels are associated with an increased incidence of CKD[39]. Thus, there is a continuum between the pre- and post-CKD onset role of sex hormones on kidney function, with androgens being risk factors and estrogens being protective[40].

We demonstrated that more candidate genes with AREs were found than expected by chance. AREs are small spanning only 14 base pair positions. Accordingly, we did not observe physical overlaps between our credible sets and AREs. However, it is still conceivable that AREs result in sex-differential gene expression due to different intensities of androgen receptor binding, resulting in sex-dependent modulations of genetic effect sizes of the regulated genes. Indeed, five of the candidate genes also showed sex-biased gene-expression in several tissues (Supplementary Data 13)[24]. Olivia et al. also demonstrated that chromosome X genes showed an enrichment of sex-biased gene expressions and eQTL sex-interactions, motivating the conduct of sex-stratified and interaction analyses for X-chromosomal variants to understand the genetics of sex dimorphisms. In particular, this also applies for kidney traits, given our observation of a significantly higher X-chromosomal heritability in males compared to females for eGFR and UA.

Our analyses are based on the assumption of complete X-inactivation in women with random selection of the inactivated chromosome. The pseudoautosomal regions were excluded from the analysis. X-inactivation is a complex, not fully understood process with several open questions e.g., regarding chromosome selection, progression, cell type dependence and stability[41]. Deviations from the above assumptions have been described such as escape from X-inactivation[42], which may contribute to sex-biased gene-expression. In case of incomplete X-inactivation, effect sizes of women are over-estimated according to our model, which could result in false positive interactions showing higher effects in females. In our case, this could affect the interactions observed at our female-specific candidate *DRP2* and the interaction at *HPRT1* showing larger effect sizes in females. Under a model assuming no inactivation, both genetic sex-interactions would be non-significant (Supplementary Data 4). However, to the best of our knowledge, these genes were not described as X-inactivation escapees[30,31].

No genome-wide significant findings were detected for UACR and BUN, the two other quantitative kidney-related traits. For BUN the low sample size could contribute to this result. Since there is no standard protocol for urine collection and because of issues in measuring urine albumin, reliability of UACR assessment could be compromised reducing statistical power to detect associations[43,44]. Likewise, the binary traits CKD, MA and gout showed no genome-wide associations, demonstrating the lower power of binary traits compared to quantitative traits. However, it needs to be acknowledged that we applied the stringent cut-off for genome-wide significance despite analyzing only chromosome X variants. We performed cross-phenotype comparisons using all traits but found only locus 1 (Xp22.31) to be nominally associated with UACR and MA. Regarding eGFR signals, 11 respectively nine nominally significant co-associations with CKD respectively BUN were observed, all with the expected directions of effects.

The fact that about 80% of the study participants were of European ancestry has limited the power to identify genetic heterogeneity across ancestries. Nevertheless, by meta-regression analysis, we identified heterogeneity at the Xq28 locus, likely related to the pronounced allele frequency differences between ancestries. We identified another UA locus at Xq13.1 showing different effect directions between African and European/Asian ancestries. The variant was also found in a study of Asian subjects[15]. Possible ethnic heterogeneity of this locus needs to be validated by other studies with larger percentages of African ancestries.

For all loci, we assigned likely candidate genes by our gene-prioritization strategy mainly considering high CADD score variants, eQTL colocalization results, and, in case of sex-interactions, AREs. Regarding the new loci, at Xq22.1, we found a missense mutation of *TSPAN6* associated with eGFR. Another member of the tetraspanin family was also found to be associated with eGFR[14]. This family of membrane proteins is ubiquitously expressed and involved in a multitude of cellular processes. Although a direct role with respect to kidney function was not yet described, the gene family was shown to be associated with immune function and could be involved in chronic inflammatory processes that contribute to CKD[45]. At the same cytoband, we detected a female-specific association, for which we assigned *DRP2*. Due to its described involvement in muscle dystrophy[29], we cannot exclude that this association may be related to muscle mass rather than kidney function. Indeed, BUN was not associated with the index variant. The locus was in close proximity but statistically independent of the known *ARMCX2* locus. At Xq22.3, we assigned *CLDN2* for its known role in nephrolithiasis development and a knock-out mouse model that showed kidney stone formation[33]. This locus is in proximity to the known *RNF128* described by others[32]. Indeed, we observed two independent variants at this locus, one supporting *RNF128* while the other is in the gene body of *CLDN2*. Since the signals are driven by low-frequency variants, further analyses are required to confirm the proposed locus heterogeneity. Finally, at Xq23 we assigned *ACSL4* as a plausible candidate based on eQTL colocalization. This gene was found to be deleted in a family with Alport syndrome[34].

Limitations of our study are the relatively small size of non-European ancestry samples, as well as for some of the kidney traits. Associated variants explained about 26% and 13% of the estimated eGFR and UA X-chromosomal heritabilities, respectively. Larger and more diverse studies are required to find further X-chromosomal variants, to analyze their sex interactions and to unravel their heterogeneity due to genetic ancestry. Moreover, other eGFR formula could be considered and other kidney function parameters with known sex-dimorphisms such as kidney function decline should be analyzed in future studies[46]. Y chromosomal markers are also understudied and should be included in future analyses. In the present study, we did not control for diabetes mellitus status. None of our index variants were in LD with a diabetes variant and only one (*AR/EDA2R* locus) was in LD with fasting insulin as a diabetes related trait.

In conclusion, we performed a comprehensive genetic association analyses of chromosome X variants regarding a variety of kidney traits. We discovered significant associations at four new loci, as well as six loci with new genetic sex interactions. Gene prioritization identified plausible candidate genes for all loci. In particular, candidate genes of loci showing SNP-sex interactions showed AREs and sex-biased gene expression, which could explain the observed interactions. These

findings contribute new insights into sex-dimorphisms and hormone dependance of kidney traits along with new prioritized gene targets for further molecular research.

## Methods

### Study design
We performed a cross-ancestry X-chromosome-wide association study of seven kidney traits namely the quantitative traits eGFR, serum UA, UACR, BUN, and the binary traits (CKD, gout, and MA). Sex-stratified and combined analyses were performed for 40 studies including up to 908,697 subjects (Supplementary Data 3) and considering up to 1,032,701 SNPs. We searched for additional SNP associations by performing meta-regression analyses considering ethnic origin. Results of eGFR were replicated in independent samples of the HUNT study ($N = 69,389$). Genome-wide significant loci were tested for sex interactions and were compared between traits. The study design is depicted in Supplementary Fig. 8.

### Collecting individual study data
Analyses are based on data collected in the framework of the CKDGen consortium[47]. A centrally designed and standardized analysis plan including scripts for phenotype definition, covariate handling, recommendations for data quality assurance and pre-processing, analyses modes requested and troubleshooting information was provided to all participating study groups. Only studies with approved local ethics votes and available written informed consent of study participants were considered. Details can be found in the corresponding publications of the autosomal analysis results[7–9].

### Quantitative phenotypes
Individual study details of measurement protocols and population distributions of kidney traits are shown in Supplementary Data 1. Phenotype definitions of eGFR and BUN are explained in detail in ref. 9. In brief, eGFR of adults was estimated based on serum creatinine using the 2009 CKD-EPI equation[48], winsorizing at 15 and 200 ml/min/1.73 m$^2$ as detailed previously. Studies of children or adolescents (age $\leq 18$ years) used a revised formula proposed by ref. 49. When blood urea but not BUN was available, BUN was calculated by dividing blood urea in mg/dl by 2.14. UA was analyzed in mg/dl. Urinary albumin values below the lower limit of detection (LOD) of the laboratory assay were set to the LOD. UACR in mg/g was obtained as [urinary albumin, mg/l]/[urinary creatinine, mg/dl]/100.

### Disease phenotypes
CKD cases and controls were identified as those individuals who had an eGFR <60 and ≥60 ml/min/1.73 m$^2$, respectively. MA cases and controls were identified as having UACR > 30 and <10 mg/g, respectively, as detailed in[7]. Gout was defined either by self-report, use of urate-lowering therapy, or by ICD codes, as described previously in detail[8].

### Genotyping
Genotyping was performed study-wise using micro-array platforms (see Supplementary Data 2 for details). Genotype calling, quality control and pre-processing was performed by each individual study, independently. Studies performed genotype imputation using either the Haplotype Reference Consortium v1.1 or the 1000 Genomes project phases 1v3 or 3v5 panels, using a variety of standard imputation software or own computational pipelines (Supplementary Data 2). Study-wise settings of the software were not collected by our consortium, but we compared standard errors of provided effect estimates for males and females at chromosome X and autosomes for obvious deviations from the expectations (details see below). In case of peculiarities, we queried the study centers. Imputed genotypes were analyzed as allele-dosages. Variants were annotated according to the NCBI build version b37.

### Single study association analyses
eGFR, UACR and BUN were logarithmized (natural logarithm) and residualized with respect to age, and untransformed values of UA were residualized with respect to age prior to association analysis. Moreover, UACR residuals were inverse normal transformed prior to genetic association analysis.

Studies performed sex-stratified analyses of X-chromosomal variants. All analyses were performed using appropriate regression models (linear regression of quantitative traits or logistic regression of binary traits) considering allele-dosages as independent predictors. Further adjustments of continuous phenotypes e.g., with respect to relatedness, ethnic principal components or study-specific covariables were left at the discretion of the single study analysts. Binary traits were also adjusted for age. Software packages used for association analyses included PLINK, SNPTEST, EPACTS and other (Supplementary Data 2). Single study summary statistics were uploaded to a server for central quality control and meta-analyses.

### Study quality control and harmonization
Single study results were quality controlled by comparing allele frequencies with those of the respective references discarding variants with >20% deviation using the R package EasyQC[50]. We also filtered variants with an imputation quality score <0.5 (e.g., MACH $r^2$ or IMPUTE info score), minor allele count <6, minor allele frequency <0.01, and SNPs within the pseudoautosomal regions. This resulted in up to 271,730 high-quality SNPs used for genetic association analysis (Supplementary Data 3).

Allele-dosages of chromosome X were harmonized across studies. Imputation software setting-specific coding of allele dosages (i.e., 0/1 vs. 0/2 for male A/B genotypes, respectively 0/0.5/1 vs. 0/1/2 for female AA/AB/BB genotypes) were identified through comparison of standard errors of X-chromosomal analyses of males and females with those of the respective autosomal analyses. This resulted in a characteristic pattern allowing the inference of X-chromosomal allele coding. Ambiguous cases were clarified with the single study analysts. All summary statistics were harmonized to a male 0/2 vs. female 0/1/2 genotype coding.

### Cross-ancestry meta-analysis of chromosome X variants
We first combined the summary statistics of males and females per study using fixed-effect inverse variance estimates. For this purpose, we harmonized the variant sets by filtering variants for which male and female allele frequencies differed by more than 20%. The genomic control factor $\lambda_{GC}$ was determined on the basis of chromosome X variants only. Genomic control correction was applied in case of $\lambda_{GC} > 1$. After combining the sexes, genomic control was applied for the single-study results of the overall analysis, if necessary.

Meta-analysis of studies was carried out for three analysis groups, overall, males and females, by summarizing their respective single-study statistics using inverse variance estimates. For the purpose of locus identification, we only considered variants for which summary statistics of at least ten studies were available. This excludes the phenotype "gout" from locus identification (Supplementary Data 3). $I^2$ statistics were used to assess heterogeneity across studies. Variants with $I^2 > 95\%$ were discarded. We also discarded variants with weighted minor allele frequency <0.02 or weighted info score <0.8. Study sample sizes served as weights. Association $p < 5 \times 10^{-8}$ were considered genome-wide significant.

### Variance explained
Explained variance of single-SNP associations was calculated using the formula $r^2 = \beta^2/(\beta^2 + N^* \text{se}(\beta)^2)$, where $\beta$ is the estimate of the fixed-effect model, $\text{se}(\beta)$ is the respective standard error, and $N$ is the sample size[51].

## Locus definition

Genome-wide significant findings were only found for eGFR and UA. Genomic loci containing genome-wide significant associations were defined separately for each trait and primarily in the overall analysis. For each trait a locus was defined as the SNP with the lowest $P$ value (index SNP) across chromosome X with a corresponding 1-Mb segment centered around this index SNP. This procedure was repeated until no further genome-wide significant SNPs remained. Since this procedure did not cover all genome-wide hits found in males for eGFR, we analogously defined loci for the eGFR male subgroup and included them into further analyses. If two loci of a trait overlapped, they were merged to one locus keeping the index SNP with the lower P value as the new index SNP of the merged locus. This step was also repeated until no overlapping loci remained for the considered trait.

## Interaction analysis

We performed genetic sex-interaction analyses of all index SNPs.[52] Thus, we calculated the differences between sex-specific meta-effect estimates and standardized it by their corresponding standard errors considering the correlation of test statistics between males and females. We determined the Spearman rank correlation of the X-chromosome-wide beta-estimates of males and females for that purpose ($\rho_{eGFR} = 0.16$, $\rho_{UA} = 0.12$). A total of 23 SNPs were tested for interaction. To have summary statistics for both sexes for all index SNPs, we did not filter variants due to low number of available studies for this purpose (minimum number of studies was eight for UA). Since escape from X-inactivation could bias interaction analyses towards larger effect sizes in females, we also performed a sensitivity analysis assuming the extreme case of no inactivation. For that purpose, beta estimates and standard errors of female effects were halved prior to interaction analysis.

We also performed colocalization analyses of male and female statistics for all loci to test for a shared underlying causal variant. Colocalization analyses were performed using the "coloc.abf" function from the R package "coloc" (available on CRAN) based on ref. 53. Bayesian posterior probabilities (PP) for the five hypotheses were computed: H0: No associations within locus, H1: Associations within males only, H2: Associations within females only, H3: Association in both sexes but different causal variant and H4: Association within both sexes with the same causal variant. We considered a posterior probability of ≥75% as sufficient support for one of the hypotheses.

## Cross-trait comparisons

All index variants were looked up in our meta-GWAS of BUN to assess potential relevance for kidney function using the same classification as in ref. 9. In brief, relevance is considered "likely" if respective BUN associations showed opposite effect directions when compared to eGFR and nominal significance in one-sided testing. Relevance is "unlikely" if significance is achieved with the same effect direction as eGFR. All other cases are classified as "inconclusive". Index variants were also tested for associations with the binary trait CKD to assess clinical relevance, and UACR and MA to assess relevance for kidney damage. We performed one-sided testing ($p < 0.05$) according to the expected directions of effects, i.e., we expected opposite effect direction for eGFR hits and the same effect direction for UA hits. We also compared the associations of eGFR and UA.

## Identification of independent variants per locus

We identified independent SNPs per locus by performing conditional analyses. A LD map was estimated on the basis of the UKBB study–the largest contributing study, as recommended[54]. For that purpose, samples were filtered for white British ancestry, complete sex and relatedness information and carrying sex chromosome configurations that are either XX or XY in agreement with the reported sex. Summary statistics of our cross-ancestry meta-analyses were used to identify independent variants since the effect of other than European ancestries on meta-analysis results was small throughout (Supplementary Fig. 9). For each locus, the GCTA function COJO SLCT[55] was applied to identify independent variants in a step-wise forward selection process. Association statistics conditional to previously selected variants were calculated using the GCTA function COJO COND. The default collinearity cut-off of 0.9 was used for all analyses. Conditional $p$-values below the genome-wide cut-off of $5 \times 10^{-8}$ were considered independently significant. Conditional analysis was performed per trait, locus and subgroups with genome-wide significant variants within the respective locus.

## Credible set analysis

To determine likely causal variants, we calculated credible sets for all independent hits[56,57]. Search was restricted to the respective locus of an independent variant. Conditional statistics were considered in case of multiple independent variants per locus. We used the R package "gtx" to calculate Approximate Bayes Factors for the variants in the locus using respective (conditional) effect estimates and standard errors. Priors for the standard deviation were estimated empirically based on the difference of the 97.5% and the 2.5% percentile of the distribution of effect sizes within the locus. Results varied in between 0.00069 and 0.00826. PP were calculated using the derived Bayes factors and were ordered to define the cut-off for 99% credibility.

## Bioinformatic annotation of variants

Variants were annotated with a number of bioinformatics resources[58]. In brief, variants were annotated by Ensembl 2018[59] based gene look-up in a region of ±250 kb around the variant, deleteriousness scores (CADD score[60] and Regulome score[61]), linkage disequilibrium (LD, $r^2 > 0.3$) with other GWAS variants according to the GWAS catalog[62] downloaded at July 19th 2022 and LD with eQTLs of the GTEx V8 catalogue (dbGaP Accession phs000424.v8.p2)[63], downloaded at June 9th 2020. LD was calculated on the basis of 1000 Genomes Phase 3, version 5 reference panel for European populations. A variant was considered unreported, if not in LD ($r^2 > 0.3$) with a variant previously reported for the respective trait.

## Colocalization analysis of gene-expression quantitative trait loci

We tested for overlapping causal variants between kidney trait associations and gene-expression quantitative trait loci (eQTLs). For this purpose, we used eQTL data from the current release of GTEx V8 and of the NephQTL database (glomerular and tubulointerstitial tissues of the kidney, NEPTUNE)[64,65]. Genome-builds of GTEx (hg38) and our GWAS (hg19) were harmonized by lifting GTEx eQTLs to hg19 using the SNP lookup table provided by GTEx (see above). For primary interpretation, we considered the following tissues: kidney cortex (primary tissue of interest), adrenal gland (due to involvement in aldosterone signaling, importance for water and salt homeostasis and production site of sex hormones), whole blood (best power due to highest number of known eQTLs) and muscle skeletal (as alternative source of serum creatinine and different metabolism in males/females) from GTEx, and, kidney glomerular and kidney tubulointerstitial from NephQTL. For each independent genome-wide significant SNP per analysis group, we considered annotated nearest genes (±250 kb window) and genes regulated in *cis* (*cis*-eQTLs with $r^2 \geq 0.3$ with the index variant). Annotation of gene symbols with Ensembl-ID for GTEx and Entrez-ID for NEPTUNE was done with an annotation table for chromosome X from HGNC[66] (downloaded November 25th 2022). For colocalization analysis, we used the intersection of available eQTLs with those analyzed in our X-chromosome-wide meta-analysis. PP ≥ 75% for H4 (shared signal) were considered as sufficient evidence for colocalization of the signals of the kidney trait and the respective gene expression. In

contrast, PP ≥ 75% for H3 (independent signals) was considered as sufficient evidence for independent signals.

### Analysis of overlap of eGFR and UA signals

To identify loci for which eGFR and UA can be traced back to the same variant, we determined positional overlap between eGFR and UA loci. We then calculated the LD ($r^2$) between the respective top-associated variants using the UKBB LD-map (see *Identification of independent variants per locus*). To be conservative with claiming independent loci, a value of $r^2 ≥ 0.1$ was considered as overlap. Moreover, we performed formal colocalization analysis between eGFR and UA signals for the merged SNP lists of both loci. Colocalization analysis was performed for the stratum displaying the lowest index p-value (male, female, overall). Again, to be conservative a PP(H4) ≥ 50% was counted as overlap for this analysis.

### Colocalization analysis with testosterone

We performed colocalization analysis of our loci with testosterone to check whether signals could be primarily driven by testosterone. We used the summary statistics of ref. 67 for that purpose. PP(H4) ≥ 75% was considered as sufficient evidence for colocalization.

### Validation analysis in HUNT study

We used the HUNT study to validate our findings in the overall analysis of eGFR. A locus was considered validated, if the top-variant of the meta-analysis was nominally significant in HUNT with the same effect direction (one-sided tests).

### Cross-ancestry meta-regression analysis of chromosome X variants

To account for mixed ethnicities of our contributing studies, we applied meta-regression analysis as implemented in the MR-MEGA package (v0.1.2)[68]. Ethnicity was accounted for by three axes of genetic variation calculated on the basis of the autosomal data. The chromosome X-wide findings of our meta-analysis were checked for ethnic heterogeneity of the effects by considering the p-value of the respective estimates ($p_{anc-het}$) and by visually inspecting Forest-Plots regarding reported study ethnicity.

### Frequency of androgen response elements and respective gene regulations

Candidate gene assignments of variants showing genetic sex-interactions were partly based on the presence of AREs according to Wilson et al.[18]. To assess how frequent this annotation occurs by chance, we randomly selected and annotated 1000 variants from our analysis. It revealed that AREs of tier three or better occurred in 49% of our SNP annotations while proven ARE induced regulation of gene-expression was found for only 4.3% of the markers. Formal enrichment analysis of actually found genes with (regulating) AREs was performed using the exact binomial test.

### Assignment of candidate genes

We used our secondary analyses and annotation of our independent variants to prioritize genes at our loci. Gene-prioritization is based on the following (ordered) criteria.

(1) Missense mutations with high CADD score (>10) in the credible set of the variant with PP > 1%.
(2) Co-localization of locus with an eQTL at a kidney tissue.
(3) Co-localization of locus with an eQTL of a gene with known relevance to kidney function (see below) in any tissue.
(4) For variants showing sex-interactions only: nearby genes of known kidney function with androgen response elements (AREs) based on Wilson et al.[18] with minimum tier three elements.
(5) Genes with known kidney function nearby high CADD score variants (>10) in the credible set of a variant.
(6) Genes with known kidney function nearby the variant.
(7) Co-localization of locus with an eQTL of any gene in any tissue.
(8) Genes nearby high CADD score variants (>10) in the credible sets of the variant.
(9) Genes nearby the variant.

At this, possible functional relationship of a gene with kidney phenotypes or diseases was assessed by searching Coremine Medical, Online Mendelian Inheritance in Man (OMIM) and Pubmed.

### X-chromosomal heritability

We used GCTA to estimate X-chromosomal heritability of eGFR and UA for the analysis groups overall, male and female. A random subset of 200,000 UKBB samples were analyzed for that purpose.

### Look-up of reported variants

We retrieved X-chromosomal SNP associations from the studies of refs. 13–15 and tested them for associations with kidney traits of our study. Trait associations reported in these studies were restricted to kidney traits analyzed in our study, creatinine levels and glomerular filtration rate. Nominal one-sided significance with the same effect direction was considered as successful replication.

### Reporting summary

Further information on research design is available in the Nature Portfolio Reporting Summary linked to this article.

## Data availability

Summary statistics for this study are publicly available at http://ckdgen.imbi.uni-freiburg.de/datasets/Scholz_2023. Further data are provided in the Supplementary Data file. Data sets used in this study are NephQTL (https://nephqtl.org/), GTEx V8 data (https://gtexportal.org/home/protectedDataAccess), the HUNT Study (https://www.ntnu.edu/hunt) and the UK Biobank (https://www.ukbiobank.ac.uk/).

## Code availability

All analysis scripts are provided at GitHub https://github.com/GenStatLeipzig/CKDGen_ChrX.

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

## Acknowledgements

This project was supported by the German Research Foundation (DFG) Project-ID 431984000 (SFB 1453 A.Köt., P.S.), 523737608 (P.S.), and 209933838 (SFB1052 M.S.) and the German Federal Ministry of Education and Research (BMBF, #01ZX1906B, project "SYMPATH", K.H.).

## Author contributions

M.S., K.H., A.P., A.Köt., P.S., and C.P. wrote the manuscript. M.S., K.H., J.P., A.Kü., H.K., A.Teu., A.Köt., P.S., and C.P. designed the study. M.S., B.O.A., H.B., B.B., A.C., J.Chal., D.I.C., C.C., R.C., J.M.G., D.F.G., P.H., C.A.H., B.H., H.H., B.I., J.B.J., W.K., M.E.K., A.Kör., H.Kr., B.K., M.Kä., T.L., M.L., P.M., N.Ma., K.M., Y.M., W.M., A.O., B.P., M.P., O.P., D.P., B.M.P., T.R., O.T.R., H.R., P.R., I.R., C.S., V.S., B.S., M.Si., H.S., K.J.S., K.S., H.St., M.St., P.Su., G.S., P.O.S., E.T., B.O.T., Y.T., J.T., A.Tö., U.V., Y.X.W., W.W., J.Wh., S.W., J.F.W., T.Y.W., M.Wo., X.S., U.T., A.M.H., N.F., and A.P. managed an individual contributing study. M.S., K.H., J.P., M.W., A.Kü., M.K.N., H.K., Y.L., A.H., M.G., S.G., M.Li., A.Ti., J.Ch., M.C., J.Wa., T.N., M.A., M.L.Big., T.B., J.Chal., D.I.C., M.LingC., M.LiC., X.C., G.D., T.L.E., A.G., F.G., D.F.G., P.H., C.H., I.M.H., J.N.H., P.J., Y.K., M.Kan., M.E.K., A.Kr., M.K., L.A.L., H.L., B.M.L., L.P.L., P.P.M., N.M., W.M., M.N., I.N., R.N., Y.O., N.P., L.M.R., H.R., K.M.R., A.R., K.J.S., G.S., B.O.T., C.T., L.F.T., J.Tr., P.V.M., V.V., C.W., T.W.W., M.Wo., A.Y.C., M.F., A.Teu., N.F., A.P., A.Köt., P.S., and C.P. performed statistical methods and analysis. M.S., K.H., J.P., M.W., A.Kü., H.K., Y.L., A.H., M.G., S.G., M.Li, J.Ch., M.C., J.Wa., T.N., X.C., A.G., S.D.G., D.F.G., P.H., M.E.K., A.Kr., Y.M., P.P.M., D.M., R.N., H.R., G.S., L.F.T., J.Tr., C.W., T.W.W., A.Y.C., and A.P. performed bioinformatics. M.S., K.H., J.P., A.Kü., M.K.N., S.G., M.Li, A.C., J.Chal., C.C., K.E., A.G., D.F.G., H.H., A.Kr., P.M., K.S., P.Su., G.S., P.O.S., B.O.T., M.Wo., M.F., U.T., A.Teu., A.P., A.Köt., P.S., and C.P. interpreted results. B.O.A., B.B., R.B., H.C., D.I.C., C.C., F.G., S.H., P.H., C.H., K.Hv., C.C.K., M.E.K., A.Kör., P.K., A.Kr., M.Kä., L.A.L., J.L., P.M., Y.M., N.M., G.M., D.M., J.c.M., B.P., M.P., D.P., O.T.R., H.R., D.R., A.R., J.I.R., V.S., M.Si., K.J.S., E.T., J.T., L.F.T., J.Tr., V.V., U.V., Y.X.W., J.F.W., M.F. and A.Teu. performed genotyping. J.P., M.W., A.Kü., M.K.N., H.K., Y.L., A.H., M.G., S.G., M.Li, A.Ti., M.C., J.Wa., T.N., N.B., M.L.Big., T.B., H.B., B.B., J.C., A.C., H.C., J.Chal., D.I.C., X.C., M.D., G.D., T.L.E., K.E., A.G., S.D.G., D.F.G., C.A.H., C.H., I.M.H., J.N.H., B.H., H.H., N.H., J.B.J., P.J., M.Kas., M.E.K., H.Kr., B.K., M.K., M.Kä., L.A.L., J.P.L., B.M.L., P.M., N.Ma., Y.M., P.P.M., N.M., G.M., D.M., J.c.M., W.M., M.N., K.N., I.N., R.N., I.O., A.O., B.P., M.P., N.P., O.P., D.P., B.M.P., T.R., L.M.R., O.T.R., H.R., K.M.R., A.R., J.I.R., I.R., V.S., N.S., B.S., M.Si., H.S., K.J.S., K.S., H.St., P.Su., G.S., P.O.S., K.D.T., B.O.T., C.T., P.V.M., V.V., U.V., J.Wh., S.W., J.F.W., T.W.W., M.Wo., X.S., A.Y.C., M.F., U.T., A.Teu., N.F., A.P., A.Köt., P.S., and C.P. critically reviewed the manuscript. H.B., J.C., A.C., J.Chal., C.C., R.C., M.D., K.D., P.H., C.A.H., H.H., N.H., K.Hv., J.B.J., M.Kas., A.Kr., M.Kä., J.P.L., N.Ma., K.M., W.M., K.N., I.O., A.O., B.P., M.P., O.P., D.P., T.P., T.R., O.T.R., I.R., V.S., N.S., M.Si., K.J.S., K.S., P.O.S., E.T., A.Ter., A.Tö., Y.X.W., J.Wh., S.W., J.F.W., T.Y.W., M.Wo., U.T., and A.Köt. recruited subjects.

## Funding

## Competing interests

D.F.G., H.H., I.O., K.S., P.Su., G.S., and U.T. are employees of deCODE/Amgen Inc. M.E.K. is employed with Synlab Holding Deutschland GmbH. W.M. is employed with Synlab Services GmbH and holds shares of Synlab Holding Deutschland GmbH. B.M.P. serves on the Steering Committee of the Yale Open Data Access Project funded by Johnson & Johnson. D.R. is currently an employee of Janssen Pharmaceuticals Inc. M.S. received funding from Pfizer Inc. for a project not related to this research. The remaining authors declare no competing interests.

## Additional information

Markus Scholz [1,2,140] ✉, Katrin Horn [1,2,140], Janne Pott [1,2], Matthias Wuttke [3,4], Andreas Kühnapfel [1,2], M. Kamal Nasr [5,6,7], Holger Kirsten [1,2], Yong Li [3], Anselm Hoppmann[3], Mathias Gorski [8,9], Sahar Ghasemi [5,6], Man Li[10], Adrienne Tin [11,12], Jin-Fang Chai[13], Massimiliano Cocca [14], Judy Wang[15], Teresa Nutile [16], Masato Akiyama [17,18], Bjørn Olav Åsvold [19,20], Nisha Bansal[21,22], Mary L. Biggs[23,24], Thibaud Boutin[25], Hermann Brenner [26,27], Ben Brumpton [19,28,29], Ralph Burkhardt [2,30], Jianwen Cai[31], Archie Campbell [32], Harry Campbell[33], John Chalmers[34], Daniel I. Chasman [35,36], Miao Ling Chee[37], Miao Li Chee[37], Xu Chen[38], Ching-Yu Cheng [37,39,40], Renata Cifkova[41,42], Martha Daviglus[43], Graciela Delgado[44], Katalin Dittrich[45], Todd L. Edwards [46,47], Karlhans Endlich [6,48], J. Michael Gaziano[49], Ayush Giri [50,51], Franco Giulianini[35], Scott D. Gordon [52], Daniel F. Gudbjartsson[53,54], Stein Hallan[55,56], Pavel Hamet [57,58], Catharina A. Hartman [59], Caroline Hayward [25], Iris M. Heid[8], Jacklyn N. Hellwege [46,47,60], Bernd Holleczek[26], Hilma Holm [53], Nina Hutri-Kähönen[61], Kristian Hveem[19], Berend Isermann [2,62], Jost B. Jonas [63,64,65,66], Peter K. Joshi[33], Yoichiro Kamatani [17,67], Masahiro Kanai [17,68,69], Mika Kastarinen[70], Chiea Chuen Khor [71,72], Wieland Kiess [2,45], Marcus E. Kleber [44,73], Antje Körner [2,45,74], Peter Kovacs[75], Alena Krajcoviechova[41], Holly Kramer [76,77], Bernhard K. Krämer[44], Mikko Kuokkanen [78,79,80], Mika Kähönen[81,82], Leslie A. Lange[83], James P. Lash [84], Terho Lehtimäki [85,86], Hengtong Li[40], Bridget M. Lin[31], Jianjun Liu [71,87], Markus Loeffler[1,2], Leo-Pekka Lyytikäinen [85,86], Patrik K. E. Magnusson [38], Nicholas G. Martin [52], Koichi Matsuda[88], Yuri Milaneschi [89], Pashupati P. Mishra[85,86], Nina Mononen[85,86], Grant W. Montgomery [90], Dennis O. Mook-Kanamori[91,92], Josyf C. Mychaleckyj [93], Winfried März[44,94,95], Matthias Nauck [6,96], Kjell Nikus[97,98], Ilja M. Nolte [99], Raymond Noordam [100], Yukinori Okada [101,102,103], Isleifur Olafsson[104], Albertine J. Oldehinkel [59], Brenda W. J. H. Penninx[89], Markus Perola[78,79], Nicola Pirastu [33,105], Ozren Polasek [106], David J. Porteous [32], Tanja Poulain[2,45], Bruce M. Psaty[107], Ton J. Rabelink [108,109], Laura M. Raffield [110], Olli T. Raitakari[111,112,113], Humaira Rasheed [19,28,114], Dermot F. Reilly [115], Kenneth M. Rice[24], Anne Richmond[25], Paul M. Ridker[35,36], Jerome I. Rotter [116], Igor Rudan [33], Charumathi Sabanayagam[37,39], Veikko Salomaa [78], Neil Schneiderman[117], Ben Schöttker [26,27], Mario Sims[118], Harold Snieder [99], Klaus J. Stark [8], Kari Stefansson [53,119], Hannah Stocker [26,27], Michael Stumvoll [75], Patrick Sulem [53], Gardar Sveinbjornsson[53], Per O. Svensson [120,121], E-Shyong Tai [13,87,122], Kent D. Taylor [116], Bamidele O. Tayo[76], Andrej Teren[2,123], Yih-Chung Tham[37,40], Joachim Thiery[2,62], Chris H. L. Thio [99], Laurent F. Thomas [19,124,125,126], Johanne Tremblay[57], Anke Tönjes[75], Peter J. van der Most [99], Veronique Vitart[25], Uwe Völker [6,127], Ya Xing Wang [64], Chaolong Wang [71,128], Wen Bin Wei[129], John B. Whitfield [52], Sarah H. Wild [130], James F. Wilson [25,33], Thomas W. Winkler [8], Tien-Yin Wong [37,39,131], Mark Woodward[34,132], Xueling Sim[13], Audrey Y. Chu[133], Mary F. Feitosa[15], Unnur Thorsteinsdottir[53,119], Adriana M. Hung [46,134], Alexander Teumer [5,6,7,135], Nora Franceschini[136], Afshin Parsa[137], Anna Köttgen [3,138,141], Pascal Schlosser [3,138,141] & Cristian Pattaro [139,141]

[1]Institute for Medical Informatics, Statistics and Epidemiology, University of Leipzig, Leipzig, Germany. [2]LIFE Research Center for Civilization Diseases, University of Leipzig, Leipzig, Germany. [3]Institute of Genetic Epidemiology, Department of Data Driven Medicine, Faculty of Medicine and Medical Center–University of Freiburg, Freiburg, Germany. [4]Department of Medicine IV - Nephrology and Primary Care, Faculty of Medicine and Medical Center, University of Freiburg, Freiburg, Germany. [5]Institute for Community Medicine, University Medicine Greifswald, Greifswald, Germany. [6]DZHK (German Center for Cardiovascular Research), partner site Greifswald, Greifswald, Germany. [7]Department of Psychiatry and Psychotherapy, University Medicine Greifswald, Greifswald, Germany. [8]Department of Genetic Epidemiology, University of Regensburg, Regensburg, Germany. [9]Department of Nephrology, University Hospital Regensburg, Regensburg, Germany. [10]Division of Nephrology and Hypertension, Department of Medicine, University of Utah, Salt Lake City, UT, USA. [11]Memory Impairment and Neurodegenerative Dementia (MIND) Center, University of Mississippi Medical Center, Jackson, MS, USA. [12]Division of Nephrology, Department of Medicine, University of Mississippi Medical Center, Jackson, MS, USA. [13]Saw Swee Hock School of Public Health, National University of

Singapore and National University Health System, Singapore, Singapore. [14]Institute for Maternal and Child Health, IRCCS 'Burlo Garofolo', Trieste, Italy. [15]Division of Statistical Genomics, Department of Genetics, Washington University School of Medicine, St. Louis, MO, USA. [16]Institute of Genetics and Biophysics 'Adriano Buzzati-Traverso'-CNR, Naples, Italy. [17]Laboratory for Statistical Analysis, RIKEN Center for Integrative Medical Sciences (IMS), Yokohama, Japan. [18]Department of Ocular Pathology and Imaging Science, Graduate School of Medical Sciences, Kyushu University, Fukuoka, Japan. [19]K. G. Jebsen Center for Genetic Epidemiology, Department of Public Health and Nursing, Faculty of Medicine and Health, Norwegian University of Science and Technology, Trondheim, Norway. [20]Department of Endocrinology, St. Olavs Hospital, Trondheim University Hospital, Trondheim, Norway. [21]Division of Nephrology, University of Washington, Seattle, WA, USA. [22]Kidney Research Institute, University of Washington, Seattle, WA, USA. [23]Cardiovascular Health Research Unit, Department of Medicine, University of Washington, Seattle, WA, USA. [24]Department of Biostatistics, University of Washington, Seattle, WA, USA. [25]Medical Research Council Human Genetics Unit, Institute of Genetics and Cancer, University of Edinburgh, Edinburgh EH4 2XU, UK. [26]Division of Clinical Epidemiology and Aging Research, German Cancer Research Center (DKFZ), Heidelberg, Germany. [27]Network Aging Research, Heidelberg University, Heidelberg, Germany. [28]MRC Integrative Epidemiology Unit, Population Health Sciences, Bristol Medical School, University of Bristol, Bristol, UK. [29]Clinic of Thoracic and Occupational Medicine, St. Olavs Hospital, Trondheim University Hospital, Trondheim, Norway. [30]Institute of Clinical Chemistry and Laboratory Medicine, University Hospital Regensburg, Regensburg, Germany. [31]Department of Biostatistics, Gillings School of Global Public Health, University of North Carolina at Chapel Hill, Chapel Hill, NC, USA. [32]Centre for Genomic and Experimental Medicine, Institute of Genetics & Cancer, University of Edinburgh, Edinburgh EH4 2XU, UK. [33]Centre for Global Health, Usher Institute, University of Edinburgh, Teviot Place, Edinburgh EH8 9AG, Scotland. [34]The George Institute for Global Health, University of New South Wales, Sydney, NSW, Australia. [35]Division of Preventive Medicine, Brigham and Women's Hospital, Boston, MA, USA. [36]Harvard Medical School, Boston, MA, USA. [37]Singapore Eye Research Institute, Singapore National Eye Center, Singapore, Singapore. [38]Department of Medical Epidemiology and Biostatistics, Karolinska Institutet, Solna, Sweden. [39]Ophthalmology and Visual Sciences Academic Clinical Program (Eye ACP), Duke-NUS Medical School, Singapore, Singapore. [40]Department of Ophthalmology, Yong Loo Lin School of Medicine, National University of Singapore and National University Health System, Singapore, Singapore. [41]Center for Cardiovascular Prevention, Charles University in Prague, First Faculty of Medicine and Thomayer University Hospital, Prague, Czech Republic. [42]Department of Medicine II, Charles University in Prague, First Faculty of Medicine, Prague, Czech Republic. [43]Institute for Minority Health Research, University of Illinois at Chicago, Chicago, IL, USA. [44]Vth Department of Medicine (Nephrology, Hypertensiology, Rheumatology, Endocrinology, Diabetology), Medical Faculty Mannheim, University of Heidelberg, Heidelberg, Germany. [45]University Hospital for Children and Adolescents, Pediatric Research Unit, Medical Faculty, University Medical Center, University of Leipzig, Leipzig, Germany. [46]Department of Veteran's Affairs, Tennessee Valley Healthcare System (626)/Vanderbilt University, Nashville, TN, USA. [47]Division of Epidemiology, Department of Medicine, Vanderbilt Genetics Institute, Vanderbilt University Medical Center, Nashville, TN, USA. [48]Department of Anatomy and Cell Biology, University Medicine Greifswald, Greifswald, Germany. [49]Massachusetts Veterans Epidemiology Research and Information Center, VA Cooperative Studies Program, VA Boston Healthcare System, Boston, MA, USA. [50]Division of Quantitative Sciences, Department of Obstetrics & Gynecology, Vanderbilt Genetics Institute, Vanderbilt Epidemiology Center, Institute for Medicine and Public Health, Vanderbilt University Medical Center, Nashville, TN, USA. [51]Biomedical Laboratory Research and Development, Tennessee Valley Healthcare System (626)/Vanderbilt University, Nashville, TN, USA. [52]QIMR Berghofer Medical Research Institute, Brisbane, QLD, Australia. [53]deCODE Genetics/Amgen, Inc., Reykjavik, Iceland. [54]Iceland School of Engineering and Natural Sciences, University of Iceland, Reykjavik, Iceland. [55]Department of Clinical and Molecular Medicine, NTNU, Norwegian University of Science and Technology, Trondheim, Norway. [56]Department of Nephrology, St. Olavs Hospital, Trondheim University Hospital, Trondheim, Norway. [57]Montreal University Hospital Research Center, CHUM, Montréal, QC, Canada. [58]Medpharmgene, Montreal, QC, Canada. [59]Interdisciplinary Centre Psychopathology and Emotion regulation (ICPE), Department of Psychiatry, University of Groningen, University Medical Center Groningen, Groningen, the Netherlands. [60]Vanderbilt Genetics Institute, Vanderbilt University Medical Center, Nashville, TN, USA. [61]Tampere Centre for Skills Training and Simulation, Faculty of Medicine and Health Technology, Tampere University, Tampere, Finland. [62]Institute for Laboratory Medicine, University of Leipzig, Leipzig, Germany. [63]Department of Ophthalmology, Medical Faculty Mannheim, University Heidelberg, Mannheim, Germany. [64]Beijing Institute of Ophthalmology, Key Laboratory of Ophthalmology and Visual Sciences, Beijing Tongren Hospital, Capital Medical University, Beijing, China. [65]Institute of Molecular and Clinical Ophthalmology, Basel, Switzerland. [66]Privatpraxis Prof Jonas und Dr Panda-Jonas, Heidelberg, Germany. [67]Laboratory of Complex Trait Genomics, Department of Computational Biology and Medical Sciences, Graduate School of Frontier Sciences, The University of Tokyo, Tokyo, Japan. [68]Program in Medical and Population Genetics, Broad Institute of Harvard and MIT, Cambridge, MA, USA. [69]Department of Biomedical Informatics, Harvard Medical School, Boston, MA, USA. [70]Finnish Medicines Agency, Kuopio, Finland. [71]Genome Institute of Singapore, Agency for Science Technology and Research, Singapore, Singapore. [72]Department of Biochemistry, Yong Loo Lin School of Medicine, National University of Singapore and National University Health System, Singapore, Singapore. [73]SYNLAB MVZ Humangenetik Mannheim, Mannheim, Germany. [74]Helmholtz Institute for Metabolic, Obesity and Vascular Research (HI-MAG) of the Helmholtz Zentrum München at the University of Leipzig and University Hospital Leipzig, Leipzig, Germany. [75]Department of Medicine, University of Leipzig, Leipzig, Germany. [76]Department of Public Health Sciences, Loyola University Chicago, Maywood, IL, USA. [77]Division of Nephrology and Hypertension, Loyola University Chicago, Chicago, IL, USA. [78]Finnish Institute for Health and Welfare, Helsinki, Finland. [79]Research Program for Clinical and Molecular Metabolism, Faculty of Medicine, University of Helsinki, Helsinki, Finland. [80]Department of Human Genetics and South Texas Diabetes and Obesity Institute, University of Texas Rio Grande Valley School of Medicine, Brownsville, TX, USA. [81]Department of Clinical Physiology, Tampere University Hospital, Tampere, Finland. [82]Department of Clinical Physiology, Finnish Cardiovascular Research Center - Tampere, Faculty of Medicine and Health Technology, Tampere University, Tampere, Finland. [83]Division of Biomedical Informatics and Personalized Medicine, School of Medicine, University of Colorado Denver-Anschutz Medical Campus, Aurora, CO, USA. [84]Department of Medicine, University of Illinois at Chicago, Chicago, IL, USA. [85]Department of Clinical Chemistry, Fimlab Laboratories, and The Wellbeing Services County of Pirkanmaa, Tampere, Finland. [86]Department of Clinical Chemistry, Finnish Cardiovascular Research Center - Tampere, Faculty of Medicine and Health Technology, Tampere University, Tampere, Finland. [87]Department of Medicine, Yong Loo Lin School of Medicine, National University of Singapore and National University Health System, Singapore, Singapore. [88]Laboratory of Clinical Genome Sequencing, Graduate School of Frontier Sciences, The University of Tokyo, Tokyo, Japan. [89]Department of Psychiatry, Amsterdam Public Health and Amsterdam Neuroscience, Amsterdam UMC/Vrije Universiteit and GGZ inGeest, Amsterdam, the Netherlands. [90]University of Queensland, Brisbane, QLD, Australia. [91]Department of Clinical Epidemiology, Leiden University Medical Center, Leiden, the Netherlands. [92]Department of Public Health and Primary Care, Leiden University Medical Center, Leiden, the Netherlands. [93]Center for Public Health Genomics, University of Virginia, Charlottesville, Charlottesville, VA, USA. [94]Clinical Institute of Medical and Chemical Laboratory Diagnostics, Medical University Graz, Graz, Austria. [95]Synlab Academy, Synlab Holding Deutschland GmbH, Augsburg, Germany. [96]Institute of Clinical Chemistry and Laboratory Medicine, University Medicine Greifswald, Greifswald, Germany. [97]Department of Cardiology, Heart Center, Tampere University Hospital, Tampere, Finland. [98]Department of Cardiology, Finnish Cardiovascular Research Center - Tampere, Faculty of Medicine and Health Technology, Tampere University, Tampere, Finland. [99]Department of Epidemiology, University of Groningen, University Medical Center Groningen, Groningen, the Netherlands. [100]Department of Internal Medicine, Section of Gerontology and Geriatrics, Leiden

University Medical Center, Leiden, the Netherlands. [101]Laboratory for Systems Genetics, RIKEN Center for Integrative Medical Sciences (IMS), Yokohama, Japan. [102]Department of Statistical Genetics, Osaka University Graduate School of Medicine, Osaka, Japan. [103]Department of Genome Informatics, Graduate School of Medicine, the University of Tokyo, Tokyo, Japan. [104]Department of Clinical Biochemistry, Landspitali University Hospital, Reykjavik, Iceland. [105]Biostatistics Unit – Population and Medical Genomics Programme, Genomics Research Centre, Human Technopole Palazzo Italia, Viale Rita Levi-Montalcini, 1, 20157 Milan, Italy. [106]University of Split Medical School, Split, Croatia. [107]Cardiovascular Health Research Unit, Department of Medicine, Department of Epidemiology, Department of Health Systems and Population Health, University of Washington, Seattle, WA, USA. [108]Department of Internal Medicine, Section of Nephrology, Leiden University Medical Center, Leiden, the Netherlands. [109]Einthoven Laboratory of Experimental Vascular Research, Leiden University Medical Center, Leiden, the Netherlands. [110]Department of Genetics, University of North Carolina, Chapel Hill, NC, USA. [111]Centre for Population Health Research, University of Turku and Turku University Hospital, Turku, Finland. [112]Department of Clinical Physiology and Nuclear Medicine, Turku University Hospital, Turku, Finland. [113]Research Center of Applied and Preventive Cardiovascular Medicine, University of Turku, Turku, Finland. [114]Division of Medicine and Laboratory Sciences, Faculty of Medicine, University of Oslo, Oslo, Norway. [115]Janssen Pharmaceuticals Inc., Titusville, NJ 08560, USA. [116]The Institute for Translational Genomics and Population Sciences, Department of Pediatrics, The Lundquist Institute for Biomedical Innovation at Harbor-UCLA Medical Center, Torrance, CA, USA. [117]Department of Psychology, University of Miami, Coral Gables, FL, USA. [118]Department of Social Medicine, Population and Public Health, University of California at Riverside School of Medicine, Riverside, CA, USA. [119]Faculty of Medicine, School of Health Sciences, University of Iceland, Reykjavik, Iceland. [120]Department of Clinical Science and Education, Södersjukhuset, Karolinska Institutet, Stockholm, Sweden. [121]Department of Cardiology, Södersjukhuset, Stockholm, Sweden. [122]Duke-NUS Medical School, Singapore, Singapore. [123]Department of Cardiology and Intensive Care Medicine, University Hospital OWL of Bielefeld University, Campus Klinikum Bielefeld, Teutoburger Straße 50, 33604 Bielefeld, Germany. [124]Department of Clinical and Molecular Medicine, Norwegian University of Science and Technology, Trondheim, Norway. [125]BioCore - Bioinformatics Core Facility, Norwegian University of Science and Technology, Trondheim, Norway. [126]Clinic of Laboratory Medicine, St.Olavs Hospital, Trondheim University Hospital, Trondheim, Norway. [127]Interfaculty Institute for Genetics and Functional Genomics, University Medicine Greifswald, Greifswald, Germany. [128]School of Public Health, Tongji Medical College, Huazhong University of Science and Technology, Wuhan, China. [129]Beijing Tongren Eye Center, Beijing Tongren Hospital, Capital Medical University, Beijing, China. [130]Usher Institute, University of Edinburgh, Teviot Place, Edinburgh EH8 9AG, Scotland. [131]Tsinghua Medicine, Tsinghua University, Beijing, China. [132]The George Institute for Global Health, School of Public Health, Imperial College London, London, UK. [133]Genetics, Merck & Co., Inc, Kenilworth, NJ, USA. [134]Vanderbilt University Medical Center, Division of Nephrology & Hypertension, Nashville, TN, USA. [135]Department of Population Medicine and Lifestyle Diseases Prevention, Medical University of Bialystok, Bialystok, Poland. [136]Department of Epidemiology, Gillings School of Global Public Health, University of North Carolina at Chapel Hill, Chapel Hill, NC, USA. [137]Division of Kidney, Urologic and Hematologic Diseases, National Institute of Diabetes and Digestive and Kidney Diseases, National Institutes of Health, Bethesda, MD, USA. [138]Department of Epidemiology, Johns Hopkins Bloomberg School of Public Health, Baltimore, MD, USA. [139]Eurac Research, Institute for Biomedicine (affiliated with the University of Lübeck), Bolzano, Italy. [140]These authors contributed equally: Markus Scholz, Katrin Horn. [141]These authors jointly supervised this work: Anna Köttgen, Pascal Schlosser and Cristian Pattaro. ✉e-mail: markus.scholz@imise.uni-leipzig.de

