## [Peer Review File · Nature Communications]

X-chromosome and kidney function: Evidence from a multi-trait genetic analysis of 908,697 individuals reveals sex-specific and sex-differential findings in genes regulated by androgen-response elementsREVIEWER COMMENTS

Reviewer #1 (Remarks to the Author):

This is a very interesting paper on the association between X chromosome and kidney traits. Sex chromosomes are difficult to study, and many investigators often exclude X chromosome (together with Y chromosome) from GWAS of complex traits. The authors should be congratulated on undertaking this comprehensive analysis on this under-explored and poorly understood part of the human genome in relation to CKD-related traits.

1. There is a well-known imbalance in X chromosome numbers between genders (with men having only one X chromosome). The authors did harmonise the results from men and women to account for inherent sex differences in allelic content of SNPs in men and women (as clarified in Methods section). I do think that this comes with certain assumptions and would benefit from a contemplation in the Discussion.

2. Can you please clarify how you handled pseudo-autosomal regions (PAR1 and PAR2) – I presume they were excluded from this analysis?

3. Colocalisation analyses with eQTL data – I understand that only four tissues from GTEx were used? Why?

4. Locus 14 - the prioritised gene (HPRT1) is of relevance to purine metabolism. From Table 1, it appears there was an association signal with both eGFR and uric acid (mapping onto two different SNPs in LD with each other). The results section focuses primarily on the association signal with eGFR, possibly because the magnitude of association. I do think that the stronger biological link is between HPRT1 and uric acid than between HPRT1 and eGFR; it would be helpful to emphasise it.

5. Prioritisation of genes is driven by the presence of androgen response-elements in their proximity. Given that and because androgens are considered potential contributors to CKD (and as such a likely explanation for certain presentations of sexual dimorphism of CKD) I think the paper would benefit from a little bit deeper emphasis on the linkage of androgens to kidney function – in the current version of the manuscript this is based only on one reference (BMC Medicine 2020). What is the evidence for associations between kidney functions and different androgens?

6. Sex differences in expression of X chromosomal genes are hypothesised to drive some of gender inequality in human health and disease. Indeed, silencing of one copy of X chromosome in female cells is never complete and a significant proportion of X genes escaping this dosage compensation mechanism are transcribed from both X copies in females. These X escapees are amongst the strongest biological candidates for the genetically driven sexual dimorphism. It would be helpful to contemplate some of the prioritised genes in this context.

7. DRP2 gene is an interesting candidate. It does seem to escape X-chromosome inactivation. From Table S13 it appears that higher levels of expression of this gene in women than men are not kidney-specific? Some X-escapees show at least partially tissue-specific pattern when it comes to the male-female

difference in expression. Can the unbalanced expression of this X chromosome gene explain the detected association between X chromosome and eGFR that seems specific to females?

8. Out of seven kidney-related traits, only two yielded associations with genetic variants of the X chromosome. Please confirm that explicitly within the abstract and the beginning of Results section.

9. “We investigated genome-wide significant index variants reported previously^{13–15} for associations with kidney-related traits” – please confirm that this statement pertains to the X chromosome variants only.

Reviewer #2 (Remarks to the Author):

This is a well conducted study of sex chromosomal variants and their associations with renal traits. The importance of the study stems from the fact that sex chromosomes are frequently ignored in large GWAS meta-analyses despite many human traits (including renal) exhibiting sexual dimorphism. Thus, this study provides the missing data for chromosome X and nicely supplements the existing studies of kidney function and related traits. The methods and the results are well described, and the analyses are of high quality. The interpretation of findings is appropriate, but some improvements could be made as suggested below. Most of my comments are relatively minor:

The is a trans-ethnic meta-analysis, but specific breakdown of the overall sample size by genetic ancestry should be provided. This is an important part missing in the introduction, and the reader should not search for this information in the supplement. Also, the term “cross-ancestry” is now favored over “trans-ethnic”.

The HUNT study was used for validation of the 14 loci associated with eGFR – what was the variance in eGFR explained by these loci in the validation cohort compared to the discovery cohort? This estimate would be less biased. Can validation results be provided for urate levels as well?

In the description of known loci and their interactions, it would be also helpful to state if each of these specific loci were associated with BUN and UA as well as serum Cr, especially for those loci with stronger effect in males that are also associated with testosterone level. One concern is that the loci increasing testosterone will have an effect on the muscle mass altering the rate of Cr production rather than clearance. This issue should be assessed in more detail since these loci may not be truly associated with renal function.

Related comment is that the discussion of pleiotropy does not include the direction of pleiotropic effects. For example, is the eGFR-decreasing allele associated with higher testosterone level at Locus 1? Conversely, is the eGFR-decreasing allele associated with lower estradiol levels at Locus 4? If these

directions are indeed correct, and these loci are not associated with BUN, perhaps these should be interpreted with more caution as potentially non-kidney function associated despite the fact that they had been previously reported.

When interrogating GWAS loci against ARE elements, it would be helpful to examine if any variants in strong LD with the index SNP physically intersect ARE. Alternatively, could also examine if any of the SNPs contained in credible sets intersect ARE. Currently, the discussion of these elements is based mainly on the proximity to the index SNP without accounting for extended LD (unless I am misinterpreting the results, in which case please clarify).

Similar to pleiotropic effects, it would be helpful to comment on the specific tissue and the direction of eQTL effects when discussing co-localized eQTLs. Are any of the signals for eGFR co-localizing with skeletal muscle or testicular eQTLs?

I would favor labeling the loci with the top prioritized candidate gene instead of an arbitrary number to increase readability (only a suggestion, I understand why some authors prefer not to do this).

Chromosome Y is not included in the analysis, it would be helpful for the authors to provide reasons as to why chromosome Y markers were excluded.

Were chromosome X markers imputed in a sex-specific manner?

How was diabetes handled in the analysis of renal traits such as eGFR and UACR?

What were the genomic inflation factors for the final chr. X meta-analysis for each of the traits? Can the authors list these and provide QQ plots in the supplement?

Table 1. Please indicate which loci are novel vs. previously reported.

Point by point response

We thank the reviewers very much for their helpful and constructive comments helping us to improve the manuscript.

Reviewer #1

Comment: This is a very interesting paper on the association between X chromosome and kidney traits. Sex chromosomes are difficult to study, and many investigators often exclude X chromosome (together with Y chromosome) from GWAS of complex traits. The authors should be congratulated on undertaking this comprehensive analysis on this under-explored and poorly understood part of the human genome in relation to CKD-related traits.

Authors reply: We thank the reviewer very much for the positive evaluation and the helpful and constructive comments. Please find below our point-by-point responses and improvements.

Comment 1: There is a well-known imbalance in X chromosome numbers between genders (with men having only one X chromosome). The authors did harmonise the results from men and women to account for inherent sex differences in allelic content of SNPs in men and women (as clarified in Methods section). I do think that this comes with certain assumptions and would benefit from a contemplation in the Discussion.

Authors reply: We thank the reviewer very much for pointing out that our analysis is based on specific assumptions regarding X-inactivation. The most relevant assumptions are that the X-inactivation is perfect and at random. It is believed that the second assumption might be true on average (not necessarily at an individual level given that the inactivation pattern is based on a few embryonal blood stem cells). Regarding the first assumption, there is considerable knowledge about genes escaping X inactivation. In our study, a violation of this assumption could result in false positive interaction effects suggesting higher effects in females. However, to the best of our knowledge, our respective candidate genes *DRP2* and *HPRT1* were not described as X-inactivation escapees.

Changes in manuscript: We added respective considerations to the discussion. For sensitivity analysis of interaction effects suggesting higher genetic effects in females, we added an interaction test assuming no inactivation (methods, new column in Table S4).

Methods (section “interaction analysis”): *Since escape from X-inactivation could bias interaction analyses towards larger effect sizes in females, we also performed a sensitivity analysis assuming the extreme case of no inactivation. For that purpose, beta estimates and standard errors of female effects were halved prior to interaction analysis.*

Discussion (5th paragraph): *In case of incomplete X inactivation, effect sizes of women are over-estimated according to our model, which could result in false positive interactions showing higher effects in females. In our case, this could affect the interactions observed at our female-specific candidate *DRP2* and the interaction at *HPRT1* showing larger effect sizes in females. Under a model assuming no inactivation, both genetic sex-interactions would be non-significant (Table S4). However, to the best of our knowledge, these genes were not described as X-inactivation escapees^{30,31}.*

Comment 2: Can you please clarify how you handled pseudo-autosomal regions (PAR1 and PAR2) – I presume they were excluded from this analysis?

Authors reply: Only 27 SNPs were in the PAR regions. We have now removed these variants from our association analysis for consistency, which does not affect the results.

Changes in manuscript: We added this filter to the methods section “study quality control and harmonization” and revised the respective tables.

Comment 3. Colocalisation analyses with eQTL data – I understand that only four tissues from GTEx were used? Why?

Authors reply: We determined eQTL colocalization for all available tissues and reported the results in supplemental table S8 to allow the readers a comprehensive review. However, we decided to restrict our general interpretation to a few tissues to reduce multiple testing burden and to improve the interpretability of results. Besides the NephQTL tissues, we considered from GTEx the following tissues: (1) kidney cortex as the primary tissue of interest; (2) adrenal gland for sex hormone production, due to its involvement in aldosterone signaling, and its role in water and salt homeostasis; (3) whole blood due to largest number of known eQTLs; and (4) muscle skeletal to control for serum creatinine production and for the different muscle metabolism of males and females.

Changes in manuscript: We state more clearly that the tissue selection was for primary interpretation purposes only and present an explanation of our selection in the methods.

Methods (section “colocalization analysis of gene-expression quantitative trait loci”): *For primary interpretation, we considered the following tissues: kidney cortex (primary tissue of interest), adrenal gland (due to involvement in aldosterone signaling, importance for water and salt homeostasis and production site of sex hormones), whole blood (best power due to highest number of known eQTLs) and muscle skeletal (as alternative source of serum creatinine and different metabolism in males/females) from GTEx, and, kidney glomerular and kidney tubulointerstitial from NephQTL.*

Comment 4: Locus 14 - the prioritised gene (HPRT1) is of relevance to purine metabolism. From Table 1, it appears there was an association signal with both eGFR and uric acid (mapping onto two different SNPs in LD with each other). The results section focuses primarily on the association signal with eGFR, possibly because the magnitude of association. I do think that the stronger biological link is between HPRT1 and uric acid than between HPRT1 and eGFR; it would be helpful to emphasise it.

Authors reply: We thank the reviewer very much for this remark. We agree that the biological link with uric acid is stronger than with eGFR. We now mention this in the revised version of the manuscript.

Changes in manuscript: In the description of Locus 14, we now state that the biological link with uric acid is stronger.

Results (section “locus 14”): *Thus, the biological link to the observed association with UA is closer than the one observed with eGFR. Rare loss-of-function variants in HPRT1 are a cause of Lesch-Nyhan Syndrome featuring highly elevated levels of UA (OMIM-ID 308000)²⁸. In consequence, HPRT1 is the most plausible candidate gene at this locus.*

Comment 5: Prioritisation of genes is driven by the presence of androgen response-elements in their proximity. Given that and because androgens are considered potential contributors to CKD (and as such a likely explanation for certain presentations of sexual dimorphism of CKD) I think the paper would benefit from a little bit deeper emphasis on the linkage of androgens to kidney function – in the current version of the manuscript this is based only on one reference (BMC Medicine 2020). What is the evidence for associations between kidney functions and different androgens?

Authors reply: We agree that the discussion in this regard was underdeveloped.

Changes in manuscript: We improved the discussion of the role of hormones in kidney function by adding a respective paragraph in the discussion section. Due to the large body of literature, we selected

a few references only. If the reviewer has a recommendation for another reference which should be considered here, we are happy to add it.

Discussion (3rd paragraph): *Several lines of evidence suggest that sex hormones may play a role in kidney function and may contribute to sexual dimorphism of CKD. Higher levels of the sex hormone binding globulin (SHBG), a modulator of several sex hormones, have been causally associated with lower CKD risk³⁵ and gout³⁶ in men but not in women. Androgens are inversely associated with kidney function in men,³⁷ with testosterone being causally associated with lower creatinine- and cystatin-based eGFR as well as increased risk of CKD and albuminuria in men¹⁷. Dihydrotestosterone may lead to dysregulation of several metabolic pathways associated with diabetes and CKD³⁸. In contrast, lower estrogen levels are associated with an increased incidence of CKD³⁹. Thus, there is a continuum between the pre- and post-CKD onset role of sex hormones on kidney function, with androgens being risk factors and estrogens being protective⁴⁰.*

Comment 6: Sex differences in expression of X chromosomal genes are hypothesised to drive some of gender inequality in human health and disease. Indeed, silencing of one copy of X chromosome in female cells is never complete and a significant proportion of X genes escaping this dosage compensation mechanism are transcribed from both X copies in females. These X escapees are amongst the strongest biological candidates for the genetically driven sexual dimorphism. It would be helpful to contemplate some of the prioritised genes in this context.

Authors reply: We agree that this issue requires further considerations. Indeed, deviations from the assumed model of total X-inactivation could result in an over-estimation of genetic effect sizes in females. Higher genetics effects in females were found for two gene-loci reported in our analysis, namely *DRP2* and *HPRT1*. We searched the literature for reported X-inactivation escapees but to our understanding *DRP2* and *HPRT1* are not affected. Please find below the searched literature. We are happy to revise our statement in case that we overlooked something.

Literature research regarding X-inactivation escape of *HPRT1* and *DRP2* showing larger genetic effect sizes in females in our analysis:

1. Wainer Katsir K, Linial M. Human genes escaping X-inactivation revealed by single cell expression data. *BMC Genomics*. 2019 Mar 12;20(1):201. doi: 10.1186/s12864-019-5507-6. PMID: 30871455; PMCID: PMC6419355. - *DRP2 and HPRT1 not explicitly mentioned*.
2. Galupa R, Heard E. X-Chromosome Inactivation: A Crossroads Between Chromosome Architecture and Gene Regulation. *Annu Rev Genet*. 2018 Nov 23;52:535-566. doi: 10.1146/annurev-genet-120116-024611. Epub 2018 Sep 26. PMID: 30256677. - *DRP2 and HPRT1 not explicitly mentioned*.
3. Navarro-Cobos MJ, et al. Genes that escape from X-chromosome inactivation: Potential contributors to Klinefelter syndrome. *Am J Med Genet C Semin Med Genet*. 2020 Jun;184(2):226-238. doi: 10.1002/ajmg.c.31800. Epub 2020 May 22. PMID: 32441398; PMCID: PMC7384012. - *DRP2 and HPRT1 not explicitly mentioned*.
4. Shvetsova E, et al. Skewed X-inactivation is common in the general female population. *Eur J Hum Genet*. 2019 Mar;27(3):455-465. doi: 10.1038/s41431-018-0291-3. Epub 2018 Dec 14. PMID: 30552425; PMCID: PMC6460563. - *DRP2 and HPRT1 not explicitly mentioned*.
5. Carrel L, Willard HF. X-inactivation profile reveals extensive variability in X-linked gene expression in females. *Nature*. 2005 Mar 17;434(7031):400-4. doi: 10.1038/nature03479. PMID: 15772666. - *DRP2 and HPRT1 not explicitly mentioned*.
6. Cotton AM, et al. Analysis of expressed SNPs identifies variable extents of expression from the human inactive X chromosome. *Genome Biol*. 2013 Nov 1;14(11):R122. doi: 10.1186/gb-2013-14-11-

r122. PMID: 24176135; PMCID: PMC4053723. - *The genes are classified as non-escapees. HPRT1: 0%, DRP2: 15% escape.*

7. Park C, et al. Strong purifying selection at genes escaping X chromosome inactivation. *Mol Biol Evol.* 2010 Nov;27(11):2446-50. doi: 10.1093/molbev/msq143. Epub 2010 Jun 9. PMID: 20534706; PMCID: PMC2981488. - *HPRT1 was not mentioned. DRP2 was classified as inactivated.*

8. Zhang Y, et al. Genes that escape X-inactivation in humans have high intraspecific variability in expression, are associated with mental impairment but are not slow evolving. *Mol Biol Evol.* 2013 Dec;30(12):2588-601. doi: 10.1093/molbev/mst148. Epub 2013 Sep 10. Erratum in: *Mol Biol Evol.* 2016 Jan;33(1):302. PMID: 24023392; PMCID: PMC3840307. - *HPRT1 and DRP2 are not mentioned in the list of escapee genes.*

9. Tukiainen T, et al. Landscape of X chromosome inactivation across human tissues. *Nature.* 2017 Oct 11;550(7675):244-248. doi: 10.1038/nature24265. Erratum in: *Nature.* 2018 Mar 7;555(7695):274. PMID: 29022598; PMCID: PMC5685192. - *DRP2 and HPRT1 not explicitly mentioned.*

Changes in manuscript: We added this aspect to the discussion.

Discussion (5th paragraph): *In case of incomplete X inactivation, effect sizes of women are over-estimated according to our model, which could result in false positive interactions showing higher effects in females. In our case, this could affect the interactions observed at our female-specific candidate DRP2 and the interaction at HPRT1 showing larger effect sizes in females. Under a model assuming no inactivation, both genetic sex-interactions would be non-significant (Table S4). However, to the best of our knowledge, these genes were not described as X-inactivation escapees^{30,31}.*

Comment 7: DRP2 gene is an interesting candidate. It does seem to escape X-chromosome inactivation. From Table S13 it appears that higher levels of expression of this gene in women than men are not kidney-specific? Some X-escapees show at least partially tissue-specific pattern when it comes to the male-female difference in expression. Can the unbalanced expression of this X chromosome gene explain the detected association between X chromosome and eGFR that seems specific to females?

Authors reply: We thank the reviewer very much for the comment. Indeed, according to Oliva et al. *Science* 2020 (PMID: 32913072), gene-expression of this gene is higher in females than in males in kidney cortex but also other tissues, but differences are relatively mild according to GTEx (see below). Despite extensive literature research, we could not find evidence that DRP2 escapes X-inactivation (see comment 6). This suggests that the gene-expression imbalance of this gene seems to be driven by different regulation rather than inactivation escape. If the reviewer is aware of any overlooked results in this regard, we would be happy to consider this in our discussion.

Regarding the question whether the imbalanced gene-expression could explain the association, we found no colocalization between DRP2 gene-expression and eGFR in females. Thus, the eGFR association seems to be not driven by gene-expression.

Changes in manuscript: We added these considerations to the corresponding results and discussion paragraphs (5th paragraph, see comment 6).

Results (section "locus 7"): *Since there is no evidence of X-inactivation escape of this gene^{30,31}, this gene-expression difference is likely caused by different regulation but it is unlikely that this explains the observed eGFR association due to lack of colocalization of gene-expression and eGFR signals at this locus (Table S8).*

Bulk tissue gene expression for DRP2 (ENSG00000102385.12)

Comment 8: Out of seven kidney-related traits, only two yielded associations with genetic variants of the X chromosome. Please confirm that explicitly within the abstract and the beginning of Results section.

Changes in manuscript: We agree and now mention this fact in the revised version of the abstract and the results section.

Results (2nd paragraph): *After processing up to 271,730 high quality single nucleotide polymorphisms (SNPs; **Table S3**), not detecting any sign of genomic inflation (**Figure S1**), in the overall analysis we identified 14 independent loci significantly associated with eGFR and seven independent loci significantly associated with UA (**Figure 1; Table 1**). None of the other phenotypes showed genome-wide significant associations.*

Comment 9: “We investigated genome-wide significant index variants reported previously^{13–15} for associations with kidney-related traits” – please confirm that this statement pertains to the X chromosome variants only.

Changes in manuscript: We regret that this is ambiguous. We confirm that only X-chromosomal variants were considered for replication and state this explicitly.

Reviewer #2 (Remarks to the Author):

Comment: This is a well conducted study of sex chromosomal variants and their associations with renal traits. The importance of the study stems from the fact that sex chromosomes are frequently ignored in large GWAS meta-analyses despite many human traits (including renal) exhibiting sexual dimorphism. Thus, this study provides the missing data for chromosome X and nicely supplements the existing studies of kidney function and related traits. The methods and the results are well described, and the analyses are of high quality. The interpretation of findings is appropriate, but some improvements could be made as suggested below. Most of my comments are relatively minor:

Authors reply: We greatly appreciate the positive assessment and the valuable, constructive feedback provided by the reviewer. Below, you will find our detailed responses and the enhancements we have made in response to your comments.

Comment 1: This is a trans-ethnic meta-analysis, but specific breakdown of the overall sample size by genetic ancestry should be provided. This is an important part missing in the introduction, and the reader should not search for this information in the supplement. Also, the term “cross-ancestry” is now favored over “trans-ethnic”.

Changes in manuscript: We added sample size by ancestry in our revised supplemental Table S3 and the introduction. We also replaced “trans-ethnic” with “cross-ancestry” throughout.

Introduction (3rd paragraph): *Here, we conducted a cross-ancestry X chromosome-wide association meta-analysis pooling results of 40 studies on up to 908,697 individuals (up to 757,070 European, 152,793 Asian, and 26,371 African ancestry individuals, depending on the trait).*

Comment 2: The HUNT study was used for validation of the 14 loci associated with eGFR – what was the variance in eGFR explained by these loci in the validation cohort compared to the discovery cohort? This estimate would be less biased. Can validation results be provided for urate levels as well?

Authors reply: We thank the reviewer very much for the comment. Indeed, explained variance estimates could be a more robust further validation criterion. It turned out that resulting values are indeed similar (0.15% in HUNT compared to 0.13% in our meta-analysis). Unfortunately, urate was not measured in HUNT, i.e. respective association statistics could not be generated.

Changes in manuscript: We added the explained variance in HUNT in our results section and compared it with our meta-analysis result.

Results (3rd paragraph): *The variants explained 0.15% of the eGFR variance in HUNT, a value similar to that found in our meta-analysis (0.13%).*

Comment 3: In the description of known loci and their interactions, it would be also helpful to state if each of these specific loci were associated with BUN and UA as well as serum Cr, especially for those loci with stronger effect in males that are also associated with testosterone level. One concern is that the loci increasing testosterone will have an effect on the muscle mass altering the rate of Cr production rather than clearance. This issue should be assessed in more detail since these loci may not be truly associated with renal function.

Authors reply: Regarding Cr, we regret that no association statistics with Cr serum levels were collected in the framework of our consortium. However, we previously showed that the genetic correlation between Cr and eGFR is equal to 1 suggesting that the vast majority of associations were shared between the two traits (Wuttke et al., Nat. Gen. 2020, Supplemental Figure 6).

To investigate the relevance of an identified locus with respect to kidney function, we compared associations of index variants between all traits (Figure 4; supplemental table S5). For 15 of the 16 eGFR loci, effects on BUN were in the opposite direction compared to those on eGFR, as expected as the two traits are inversely correlated with respect to kidney function involvement, even though sometimes results were not significant. Associations for which BUN was not significant or with the same effect direction were conservatively discussed regarding kidney function (see supplement material discussion of all loci).

Regarding testosterone, we have now added a colocalization analysis to support interpretation of the observed genetic sex interactions. Indeed, locus 1 showed colocalization in males between eGFR and testosterone, suggesting that the observed effect might be driven by testosterone. None of the other loci showed colocalizations with testosterone.

Changes in manuscript: We have now added the direction of BUN association to all eGFR loci (Figure 4, Table S5a, all paragraphs discussing loci including supplement material). We performed testosterone colocalization analysis with the data of Ruth et al. Nat Med. 2020 (PMID 32042192). This was added to the methods. Results are presented in new supplemental table S14. We added respective interpretations in the results section.

Methods (new paragraph): *Colocalization analysis with testosterone: We performed colocalization analysis of our loci with testosterone to check whether signals could be primarily driven by testosterone. We used the summary statistics of Ruth et al.⁶⁷ for that purpose. $PP(H4) \geq 75\%$ was considered as sufficient evidence for colocalization.*

Results (section “locus 1”): *Moreover, we observed colocalization between eGFR and testosterone associations at this locus in males ($PP(H4)=99\%$) with opposite effect directions, i.e. the eGFR association could be driven by a primary testosterone effect (Table S14). The nearest candidate gene is FAM9B, which has an ARE 70 kb upstream of its transcription start side (TSS)¹⁸.*

Comment 4: Related comment is that the discussion of pleiotropy does not include the direction of pleiotropic effects. For example, is the eGFR-decreasing allele associated with higher testosterone level at Locus 1? Conversely, is the eGFR-decreasing allele associated with lower estradiol levels at Locus 4? If these directions are indeed correct, and these loci are not associated with BUN, perhaps these should be interpreted with more caution as potentially non-kidney function associated despite the fact that they had been previously reported.

Authors reply: We confirm both assumptions of the reviewer and agree that this information is important for interpretation. Regarding Locus 1, we also observed a colocalization with testosterone (see response to previous comment), implicating that locus 1 association might be driven by testosterone.

Changes in manuscript: We added this information.

Results (section “locus 1”): *Moreover, we observed colocalization between eGFR and testosterone associations at this locus in males ($PP(H4)=99\%$) with opposite effect directions, i.e. the eGFR association could be driven by a primary testosterone effect (Table S14).*

Results (section “locus 4”): *Other GWAS traits associated at this locus comprise sex hormone-binding globulin levels, male-pattern baldness, fasting insulin, estradiol levels with the same effect direction and prostate cancer risk.*

Comment 5: When interrogating GWAS loci against ARE elements, it would be helpful to examine if any variants in strong LD with the index SNP physically intersect ARE. Alternatively, could also examine

if any of the SNPs contained in credible sets intersect ARE. Currently, the discussion of these elements is based mainly on the proximity to the index SNP without accounting for extended LD (unless I am misinterpreting the results, in which case please clarify).

Authors reply: The reviewer is right in the assumption that we discussed AREs just by proximity to candidate genes. According to our biological interpretation, this physical proximity could be sufficient to explain genetic sex-interaction since respective genes are regulated by hormones which could modify genetic effect sizes. We performed the requested analysis and observed that there are no direct physical overlaps of our credible sets and AREs. It needs to be pointed out that AREs are very small (14bp) making direct physical overlaps unlikely.

Changes in manuscript: We now mention that there are no direct physical overlaps between credible set variants and AREs and discuss this issue more extensively.

Discussion (4th paragraph): *We demonstrated that more candidate genes with AREs were found than expected by chance. AREs are small spanning only 14 base pair positions. Accordingly, we did not observe physical overlaps between our credible sets and AREs. However, it is still conceivable that AREs result in sex-differential gene expression due to different intensities of androgen receptor binding, resulting in sex-dependent modulations of genetic effect sizes of the regulated genes.*

Comment 6: Similar to pleiotropic effects, it would be helpful to comment on the specific tissue and the direction of eQTL effects when discussing co-localized eQTLs. Are any of the signals for eGFR co-localizing with skeletal muscle or testicular eQTLs?

Authors reply: We agree that provision of effect directions would be useful to facilitate interpretation. Accordingly, locus 8/19 shows colocalization of our UA male signal and a muscle/skeletal eQTL of *TCEAL3* with the same direction of effect. Since this is a known locus, we added this observation to the supplement material discussing all loci.

Regarding testicular eQTLs, we believe that direct colocalization with testosterone signals might be more relevant for interpretation (see comment 3) and added this analysis.

Changes in manuscript: We add concordance or discordance of effect directions of all positive colocalization results at supplemental table S15 and at figure 5. We added the consideration of effect directions to the discussion of loci where appropriate.

We added a colocalization analysis with testosterone to support the interpretation (supplemental table S14, see also comment 3).

Comment 7: I would favor labeling the loci with the top prioritized candidate gene instead of an arbitrary number to increase readability (only a suggestion, I understand why some authors prefer not to do this).

Authors reply: We agree that direct labelling of loci by candidate genes provides some advantages. By our presentation we first aimed at identifying and characterizing the genomic loci by statistical means based on our own data. Second, gene assignment was performed in accordance to our prioritization strategy which is mainly based on external data. We respectfully would like to defend this kind of presentation separating the primary data analyses from the more uncertain / subjective task of gene assignment which depends on changing external resources. However, if the reviewer insists on changing this issue, we are happy to do so.

Changes in manuscript: None.

Comment 8: Chromosome Y is not included in the analysis, it would be helpful for the authors to provide reasons as to why chromosome Y markers were excluded.

Authors reply: We agree that Y chromosomal markers are also understudied. Unfortunately, we did not collect chromosome Y from our participating studies.

Changes in manuscript: In the discussion (9th paragraph), we now mention chromosome Y analyses as a valuable future work.

Comment 9: Were chromosome X markers imputed in a sex-specific manner?

Authors reply: The single study analysts were requested to performed state-of-the-art chromosome X imputation. We compared the standard errors between sexes and between chromosome X and autosomes to detect any obvious deviations from the expectations.

Changes in manuscript: We add in the methods that the study-specific settings of the imputation frameworks were checked for plausibility of the provided summary statistics by comparing standard errors between sexes and between chromosome X and autosomes.

Methods (section “genotyping”): *Studies performed genotype imputation using either the Haplotype Reference Consortium v1.1 or the 1000 Genomes project phases 1v3 or 3v5 panels, using a variety of standard imputation software or own computational pipelines (Table S2). Study-wise settings of the software were not collected by our consortium, but we compared standard errors of provided effect estimates for males and females at chromosome X and autosomes for obvious deviations from the expectations (details see below). In case of peculiarities, we queried the study centers.*

Comment 10: How was diabetes handled in the analysis of renal traits such as eGFR and UACR?

Authors reply: In previous analyses of our consortium, we could not detect any evidence of SNP x diabetes interactions regarding kidney parameters (Pattaro et al., PLoS Genet 2012; Wuttke et al., Nat Genet 2019) with UACR as a possible exception (Teumer et al., Nat Commun 2019). Therefore, for the current analysis round of our consortium, we did not ask participating studies for diabetes-stratified analysis results of chromosome X variants or adjustment for diabetes status. Confirming previous evidence, our look-up of the GWAS catalog (Table S7) did not detect any diabetes hits in LD with our index variants. The only (partial) exception was the index variant at the AR/EDA2R locus, which was in LD with a fasting insulin associated variant.

Changes in manuscript: We have now added the information of an LD variant with fasting insulin to the description of the respective locus.

Results (section “locus 4”): *Other GWAS traits associated at this locus comprise sex hormone-binding globulin levels, male-pattern baldness, fasting insulin, estradiol levels with the same effect direction and prostate cancer risk.*

Discussion (9th paragraph): *In the present study, we did not control for diabetes mellitus status. None of our index variants were in LD with a diabetes variant and only one (AR/EDA2R locus) was in LD with fasting insulin as a diabetes related trait.*

Comment 11: What were the genomic inflation factors for the final chr. X meta-analysis for each of the traits? Can the authors list these and provide QQ plots in the supplement?

Authors reply: We have now added QQ-plots and the respective lambda values as new supplemental Figure S1. We confirm that there was no genomic inflation. Lambda values are also provided in supplemental Table S3.

Changes in manuscript: We added new supplemental Figure S1 showing the QQ plots and refer to it in the results (2nd paragraph).

Comment 12: Table 1. Please indicate which loci are novel vs. previously reported.

Changes in manuscript: We now mark novel loci and respective candidate genes in bold.

REVIEWERS' COMMENTS

Reviewer #1 (Remarks to the Author):

The authors have been responsive to my comments and the manuscript has been improved by this revision. I only have one minor comment remaining: On Page 13 it reads: "The pseudoautosomal regions were therefore discarded". I am not sure that the context provided is the key reason why PARs should have been excluded from the analysis. I suggest that this sentence is re-phrased e.g. "The pseudoautosomal regions were excluded from the analysis".

Reviewer #2 (Remarks to the Author):

The authors have adequately addressed my comments.